# TOUCAN: Synthesizing 1.5M Tool-Agentic Data from Real-World MCP Environments

## Abstract

Large Language Model (LLM) agents are rapidly emerging as powerful systems for automating tasks across domains. Yet progress in the open-source community is constrained by the lack of high quality permissively licensed tool-agentic training data. Existing datasets are often limited in diversity, realism, and complexity, particularly regarding multi-tool and multi-turn interactions. To address this gap, we introduce TOUCAN , the largest publicly available tool-agentic dataset to date, containing 1.5 million trajectories synthesized from nearly 500 real-world Model Context Protocols (MCPs). Unlike prior work, TOUCAN leverages authentic MCP environments to generate diverse, realistic, and challenging tasks with trajectories involving real tool execution. Our pipeline first produces a broad spectrum of tool-use queries using five distinct models, applies model-based quality filtering, and then generates agentic trajectories with three teacher models using two agentic frameworks. Rigorous rule-based and model-based validation ensures high-quality outputs. We also introduce three extension mechanisms to further diversify tasks and simulate multi-turn conversations. Models fine-tuned on TOUCAN outperform larger closed-source counterparts on the BFCL V3 benchmark and establish a new Pareto optimum on MCP-Universe Bench.

## 1 Introduction

Large language models (LLMs) have become integral to AI applications, with LLM agents emerging as powerful systems for automating complex tasks across diverse domains Li et al. (2024). There is growing excitement about the potential of LLM agents to unlock new levels of automation across industries (Ferrag et al., 2025; Bousetouane, 2025). These agents handle multi-step workflows that require discovering the right tools from potentially large toolsets, calling them correctly with appropriate parameters, handle tool failures gracefully, and synthesizing results into accurate, context-aware responses Xu et al. (2025a). Recent advancements, such as the Model Context Protocol (MCP) (Anthropic, 2025), have streamlined tool integration by providing standardized interfaces, enabling seamless connections between LLMs and real-world environments and simplifying the process for LLM agents to discover, invoke, and execute external tools.

Despite these advancements, progress in the open-source community is constrained by the lack of high-quality, permissively licensed **tool-agentic data** for training more capable agentic LLMs. An instance of tool-agentic data comprises a task-trajectory pair, where trajectories capture sequences of planning, tool calls, tool responses, and the final model response. While previous efforts (Qin et al., 2023; Liu et al., 2024; 2025a; Prabhakar et al., 2025) have introduced datasets covering various tool-calling scenarios, they suffer from several limitations: restricted tool diversity, lack of authentic tool responses, focus on single-turn conversations between users and models, or insufficient scale, all of which constrain effective training of agentic capabilities. There is an urgent need for comprehensive, high-quality datasets that capture the full spectrum of tool-agentic interactions observed in production environments.

In this work, we bridge this gap by introducing TOUCAN, the largest publicly available tool-agentic dataset to date, comprising 1.5 million trajectories synthesized from nearly 500 real-world MCP servers. Unlike prior approaches that rely on simulated or limited toolsets, TOUCAN leverages authentic MCP environments with more than 2,000 tools to generate diverse, realistic, and challenging tasks spanning parallel and multi-step tool calls, as well as multi-turn conversations. Our pipeline

Table 1: TOUCAN comparison to open-source tool-agentic datasets. Comparison comprises total trajectories, tool calling scenarios ([S]ingle, [P]arallel, [M]ulti[S]tep) including no-tool-use edge case (irrelevance[IR]), number of multi-turn conversations, and other details about data generation. Note − indicates information not publicly available.

| Dataset | Trajectories | Tool-Call Scenarios | Multi Turn | Tool Specs | Tool Response |
|---|---|---|---|---|---|
| APIGent-MT-5K (Prabhakar et al., 2025) | 5,000 | S P MS IR | 5,000 | From $\tau$-Bench | Executed |
| ToolACE (Liu et al., 2025a) | 11,300 | S P MS IR | 509 | Synthetic | Simulated |
| Hermes Function-Calling V1 (interstellarninja) | 11,570 | S P MS IR | 1,890 | Synthetic | Executed |
| Nemotron (Tools) (Nathawani et al., 2025) | 310,051 | S P MS – | 199,610 | – | – |
| TOUCAN (This Work) | 1,527,259 | S P MS IR | 567,262 | Real | Executed |

begins by producing a broad spectrum of tool-use tasks using five distinct models with MCP server specifications, followed by model-based quality filtering to ensure relevance and difficulty. We then generate agentic trajectories with three teacher models, incorporating rigorous rule-based and model-based checks for high-quality outputs, including verification of tool execution and response accuracy. Our pipeline also integrates extensions to generate additional tasks targeting edge case scenarios, interactive conversations, and multi-turn dialogues.

Our experiments demonstrate the effectiveness of TOUCAN in enhancing LLM agentic capabilities. Models fine-tuned on TOUCAN surpass closed-source counterparts on the BFCL V3 benchmark (Patil et al., 2025), achieving superior performance in function calling accuracy across single-turn and multi-turn scenarios. Furthermore, they show substantial improvements on $\tau$-Bench (Yao et al., 2024) and $\tau^2$-Bench (Barres et al., 2025), with gains in tool selection, execution fidelity, and multi-turn reasoning under dynamic user interactions. On the recent MCP-Universe benchmark (Luo et al., 2025), which evaluates LLMs on 231 realistic tasks using 11 real-world MCP servers, TOU-CAN-tuned models achieve state-of-the-art performance within their parameter class, consistently outperforming leading models of comparable size. In summary, the contributions of our work are:

- **TOUCAN Dataset**. The largest open-source tool-agent training dataset, covering parallel and multi-step tool calls, multi-turn dialogues, and edge-case tool use. Recent reports on frontier LLM development, such as Kimi-K2 (Team et al., 2025b) and GLM-4.5 (Team et al., 2025a), highlight the value of large-scale trajectories with broad domain coverage, and TOUCAN provides an open-source alternative that bridges this gap.
- **TOUCAN Pipeline**. A pipeline that leverages any MCP specifications to generate diverse tool-agent trajectories, supports tool execution through MCP servers, and can be seamlessly extended to new tools via the MCP standard.
- **TOUCAN Checkpoints**. Our experiments demonstrate that models fine-tuned on TOUCAN mixtures surpass closed-source counterparts on the BFCL V3 and MCP-Universe benchmarks.

## 2 RELATED WORK

**The past: Tool-calling datasets and benchmarks for LLMs.** Early tool-calling datasets enabled LLMs to interact with tools like REST APIs and ML functions. The Gorilla project (Patil et al., 2023) demonstrated that fine-tuning on such data enhances tool-use over vanilla models, introducing the BFCL benchmark (Patil et al., 2025) as a standard. ToolAlpaca (Tang et al., 2023) offered cost-effective synthetic data with lower quality, while ToolLLM (Qin et al., 2023) expanded to 16,000+ APIs across domains. API Pack (Guo et al., 2025a) added cross-language diversity (Python, Java, C++), and API Blend (Basu et al., 2024) optimized dataset mixtures for robustness, laying the foundation for tool-agent advancements. More recently, APIGen has focused on domain diversification, contributing a training dataset covering 21 domains Liu et al. (2024).

**The present: Tool-calling benchmarks and datasets for LLM-agents.** Recent research has shifted toward training LLM agents for effective tool use, exemplified by models like Kimi-K2 (Team et al., 2025b) and GLM-4.5 (Team et al., 2025a), with performance assessed via benchmarks such as BFCL (Patil et al., 2025), $\tau$-Bench (Yao et al., 2024), and ACEBench (Chen et al., 2025). BFCL covers diverse scenarios including parallel, multi-step, and multi-turn tool use, while $\tau$-Bench

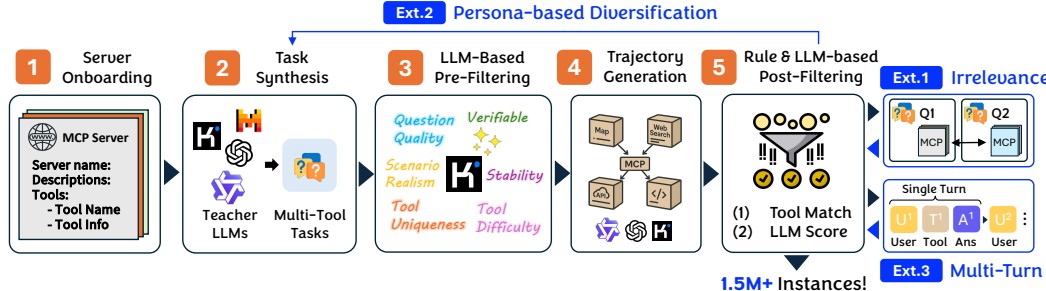

Figure 2: The TOUCAN construction pipeline: A systematic five-stage process from MCP server on-boarding through trajectory filtering, with three extensions for enhancing data diversity and realism.

focuses on realistic user-agent-tool interactions. ACEBench enhances evaluation by addressing edge cases and including a subset for tool-agent trajectories. Despite these advances, open-source training data for tool-agent trajectories remains limited. Existing datasets (interstellarninja; Liu et al., 2025a; Prabhakar et al., 2025; Nathawani et al., 2025) either lack dataset curation transparency, are small in size for SFT, simulate tool responses via LLMs, or focus on VLMs rather than LLMs Gao et al. (2025b). Table 1 compares existing tool-agentic datasets for LLMs with TOUCAN, which, at 1.5 million trajectories, offers the largest dataset, featuring extensive multi-turn dialogues, all tool-use scenarios, critical edge cases, and authentic tool responses from real-world environments.

**The future: MCP benchmarks and datasets.** As concurrent work, recent MCP benchmarks (Gao et al., 2025a; Wang et al., 2025; Luo et al., 2025; Team, 2025a; Guo et al., 2025b; Yin et al., 2025; Liu et al., 2025b; Yan et al., 2025; Team, 2025b) aim to rigorously assess LLMs in tool-use settings beyond simple correctness. For instance, MCP-Radar (Gao et al., 2025a) employs a five-dimensional evaluation including accuracy, tool selection efficiency, resource usage, parameter construction, and execution speed across software engineering, math, and problem-solving tasks with 300 queries and 42 MCP servers. Similarly, MCP-Bench (Wang et al., 2025) evaluates multi-step reasoning over 28 MCP servers and 250 tools, while MCP-Universe (Luo et al., 2025) focuses on execution-based metrics in six real-world domains. These advancements underscore the need for comprehensive training datasets to support the development of robust, open-source LLM agents.

## 3 TOUCAN: SCALING TOOL-AGENTIC DATA WITH REAL WORLD MCPS

### 3.1 TOUCAN GENERATION PIPELINE

TOUCAN is a comprehensive dataset comprising over 1.5 million tool-agent trajectories constructed using real-world tools from MCP servers. Each instance in our dataset contains a task description, a complete agent trajectory with its associated tools, quality and classification annotations, as well as comprehensive metadata. Appendix A provides a detailed schema description and demonstration samples. The construction of TOUCAN follows a systematic five-stage pipeline: MCP server onboarding, task synthesis, task filtering, trajectory generation, and trajectory filtering. Additionally, we implement three extension mechanisms to further enhance data diversity and realism. Figure 2 illustrates the complete construction pipeline. We detail each stage below.

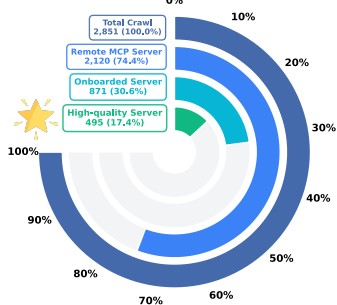

Figure 1: MCP servers filtering process

**Stage 1: MCP Server Onboarding.** To generate questions from diverse environments, the initial step involves onboarding as many high-quality MCP servers as possible. We sourced MCP server specification files from GitHub and Smithery [1], a platform and registry for MCP servers that encapsulate modular execution environments. Each MCP server is accompanied by a structured JSON document detailing metadata about the server with a machine-readable definition of the tools it provides. From an initial crawl yielding approximately 2,800 MCP servers, we applied two key filtering

---
[1]https://smithery.ai/

criteria: (1) retaining only remote MCP servers accessible via streamable HTTP to ensure compatibility with trajectory generation, and (2) excluding servers requiring third-party credentials (e.g., API keys) for tool invocation to maintain accessibility and reproducibility. This process reduced the dataset to 30.6% (871 servers). As a final step, we generated a small subset of test questions to evaluate each tool within the MCP servers, subsequently filtering out servers with problematic tools that returned error messages or failed to function correctly. This rigorous curation process resulted in a refined set of 495 high-quality MCP servers spanning diverse domains and functionalities. Figure 1 depicts the number of MCP servers retained at each filtering stage. Figure 3 demonstrates the domain distribution of the final server collection across diverse categories. The domain distribution is annotated by LLMs, where prompts can be found in Appendix D.1.

**Stage 2: Task Synthesis.** The next step involves synthesizing high-quality tasks from MCP servers, where each task comprises a question and the desired tool names from the MCP servers. The key challenge is ensuring that tasks are challenging, realistic, and cover edge cases. Therefore, we design diverse sampling strategies based on MCP server usage number from Smithery and server functionalities. To avoid potential bias from individual models, we utilized five open-source LLMs (`Mistral-Small`, `DevStral-Small`, `GPT-OSS`, `Kimi-K2`, and `Qwen3-32B`) as task generators to construct synthetic tasks (see the prompts in Appendix D.2). We apply the following three strategies to synthesize tasks, where the maximum number of tools is set to $N = 3$ in our experiments:

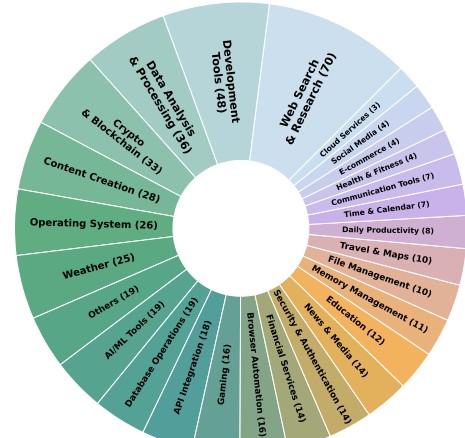

Figure 3: MCP servers distribution by domain, covering a wide range of categories. Values in parentheses indicate the number of servers belonging to each category.

**Single Server:** For a given MCP server, we synthesize tasks requiring the use of 1 to $N$ tools, ensuring a balanced selection distribution guided by server usage statistics to reflect real-world applicability.

**Multi-Server:** Leveraging LLM-based domain annotations derived from MCP metadata, we first sample $N$ MCP servers from either the same or different categories. We then prompt LLMs to conduct a server analysis, outlining potential workflows that integrate tools across these servers, targeting two to $N$ specific tools, and subsequently generating tasks that leverage functionalities from multiple servers.

**Featured Server:** Based on the original MCP file metadata, we manually selected 25 representative MCP servers from various domains, with the complete list available in Appendix B.1. In this approach, we provide all MCP server metadata within the context, specify an expected number of tools, and allow the LLM to freely explore combinations, devise realistic scenarios, select the necessary tools, and create comprehensive tasks.

**Stage 3: Task Filtering.** To ensure the quality of synthesized tasks, this stage involves annotating tasks across six dimensions and filter out suboptimal instances. We employed the `Kimi-K2` model as the annotator, which was selected for its optimal balance between correlation with human annotations and cost efficiency. The correlation statistics are detailed in Appendix C.1, and the prompt template is provided in Appendix D.4. Each dimension is rated on a 1-5 Likert scale. The detailed evaluation metrics are as follows:

- *Tool Selection Difficulty:* Judges the difficulty of selecting the required tools from provided tools.
- *Tool Selection Uniqueness:* Assesses the uniqueness of the selected tool combination relative to the available tools, and whether viable alternatives could also solve the task.
- *Question Quality:* The task's overall quality, reflected by its clarity, specificity, and effectiveness.
- *Scenario Realism:* Evaluates the authenticity and realism of the task scenario.
- *Verifiable:* Evaluates how easily the final model answer can be verified given the question.
- *Stability:* Evaluates whether tool outputs remain consistent over time, across geolocation, and under stochastic variation.

**Stage 4: Trajectory Generation.** This step involves collecting trajectories including tool calls, tool responses, and reasoning steps in agentic environments given tasks synthesized and filtered from the previous steps. To ensure diversity, we employed three LLMs from different families (GPT-OSS-120B, Kimi-K2, and Qwen3-32B) in combination with two agent frameworks (Qwen-agent and OpenAI-agent) to produce high-quality agentic trajectories. The models are deployed remotely and accessed by the agent frameworks via streamable HTTP.

**Stage 5: Rule&LLM-Based Post-Filtering.** The trajectory filtering process combines rule-based verifiers with LLM-driven annotations to ensure high quality. Rule-based heuristics exclude trajectories that fail to start the agent or connect successfully with remote MCP servers, do not contain tool calls, exhibit failures in all tool responses, or contain local file system paths. We also validate whether the trajectory uses the required tools specified by the task in the correct sequence, and report both the *desired tool use percentage* (coverage of required tools) and *order correctness* (adherence to expected sequence) metrics. We then employ GPT-OSS-120B as a judge to annotate each trajectory in terms of completeness and conciseness. The annotation prompt is provided in Appendix D.5, with metric definitions as follows:

- *Completeness:* Judges whether the assistant fulfills the user's request end-to-end.
- *Conciseness:* Judges whether the task is solved with the minimum necessary steps and verbosity.

This dual-stage filtering approach ensures that only high-quality, concise, and executable trajectories are retained in the final dataset.

## 3.2 TOUCAN EXTENSIONS

While the core pipeline generates high-quality trajectories, these are single-turn interactions between user and agent without follow-ups, which limits their practical applicability to real-world scenarios. In addition, since all available tools are contextually relevant, tool selection becomes trivial for LLMs, resulting in relatively low difficulty. To address these limitations and enhance the dataset's versatility, we apply three distinct procedures post-core pipeline (Steps 1 to 5) to generate new instances targeting specific objectives.

**Ext.1: Irrelevance.** To reduce hallucination, it is critical to train models to reject unanswerable queries or seek alternative solutions when desired tools are unavailable. To achieve this, we systematically generate queries unsolvable with the current toolset (Ext1 in Figure 2) by shuffling MCP server metadata across instances and repeating the task generation step.

**Ext.2: Persona-based Diversification.** We implement persona-based diversification (Ext2 in Figure 2) to create varied task versions. This involves two strategies: one enhances diversity by introducing new contexts and personas, while the other increases task complexity through additional constraints, all while utilizing the same target tools. This diversification process produces tasks similar yet distinct from those in the core pipeline. The prompts are detailed in Appendix D.3.

**Ext.3: Multi-Turn.** Recognizing that real-world user-agent-tool interactions seldom conform to single-turn conversations Yao et al. (2024), we introduce a self-simulation pipeline to generate multi-turn dialogues using the trajectory generation model. This is achieved through two methods: (1) splitting complex tasks requiring multi-tool coordination into sequential sub-questions, and (2) extending existing conversations by providing LLMs with context to formulate follow-up queries.

Finally, we repeat the core pipeline from steps 2 to 5 to build full trajectories with the new tasks. In the case of irrelevant tasks (Ext.1), we tighten trajectory filters to retain only instances with zero tool calls. Together, these data extensions yield a more realistic and robust TOUCAN dataset that covers all relevant tool-use scenarios and user question styles.

## 3.3 DATA ANALYSIS

This section analyzes the generated TOUCAN dataset from statistical analysis and LLM-based quality assessment.

**Statistical Analysis of TOUCAN .** We conduct comprehensive statistical analysis of MCP servers and data instances. The top MCP servers used in TOUCAN and tool statistics within each MCP servers are presented in Appendix B.2. Figure 4 provides a comprehensive analysis of the TOU-

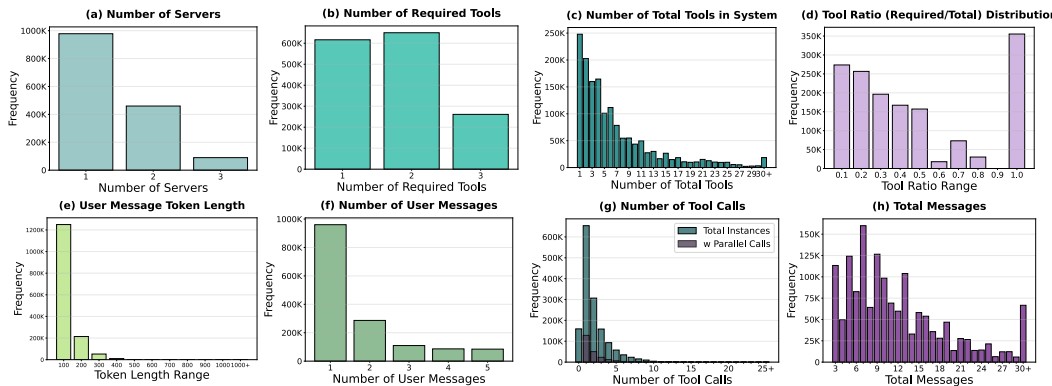

Figure 4: The figures above illustrate the TOUCAN dataset analysis. Subfigure (a) and (b) provide statistics on the number of servers and required tools per instance, highlighting TOUCAN 's comprehensive coverage of multi-server and multi-tool tasks. Subfigures (c) and (d) reveal that most tasks include more tools in the context than the targeted tools, underscoring the non-trivial tool selection challenges. Subfigure (e) displays the length of user messages in tokens. Subfigures (f) and (h) demonstrate the multi-turn nature of the tasks, characterized by extended and diverse interactions among users, agents, and tools. Subfigure (g) demonstrates that TOUCAN encompasses both single and parallel tool calls, which enhance the dataset's versatility in capturing diverse agent-tool interaction patterns.

CAN dataset. We observe that TOUCAN provides comprehensive coverage of multi-server and multi-tool tasks, and includes multi-turn conversations among users, agents, and tools. Additionally, most tasks contain more tools in the context than the required target tools, indicating non-trivial tool selection requirements. Figure 5 presents the subset statistics of TOUCAN across different trajectory generator LLMs and data partitions. We also provide embedding visualization of TOUCAN using UMAP projection in Appendix B.3, demonstrating the wide domain coverage of TOUCAN.

**Quality Assessment of TOUCAN.** Figure 6 presents a statistical analysis conducted by an LLM-as-a-judge on TOUCAN. From the task perspective (labels in 🟨), we observe that the majority of tasks exhibit exceptionally high question quality and scenario realism, indicating robust task design and alignment with real-world applications. Additionally, the dataset features a mixed difficulty range, encompassing both simple and challenging tasks. From the response perspective (label in 🟦), we find that trajectory quality is satisfactory, with most scores at or above 3 (medium) across both completeness and conciseness metrics.

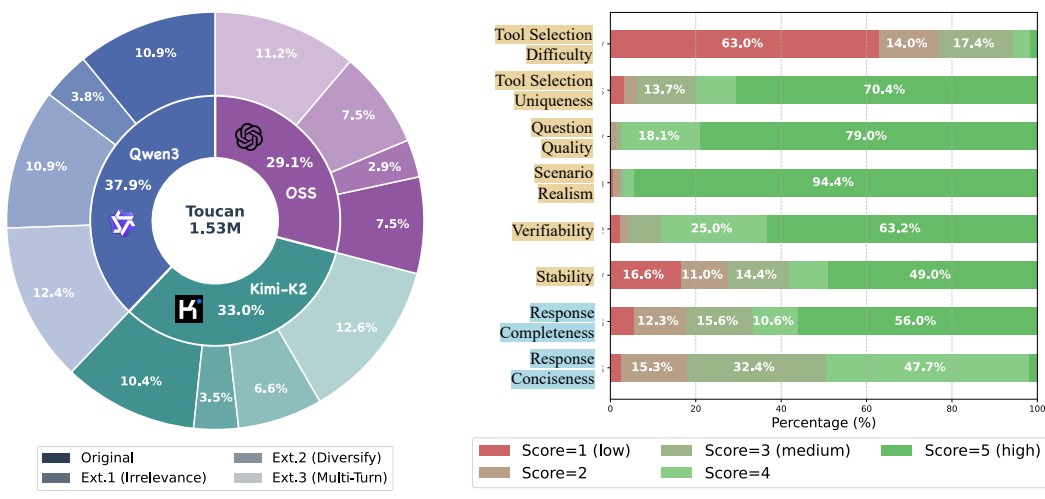

Figure 5: TOUCAN Subset Statistics

Figure 6: TOUCAN Quality Statistics

Table 2: This table compares the performance of TOUCAN -tuned models and baselines on the BFCL-V3 benchmark. We observe that TOUCAN remarkably improves baseline model performance through supervised fine-tuning (SFT) and enables smaller models to outperform larger models across different evaluation aspects.

| Model | Overall | Single Turn | | Multi Turn | Hallucination | |
| --- | --- | --- | --- | --- | --- | --- |
| | | *Non-live (AST)* | *Live (AST)* | | *Relevance* | *Irrelevance* |
| DeepSeek-V3 | 64.71% | 88.54% | 77.34% | 29.87% | **83.33%** | 76.49% |
| Qwen2.5-72B-Instruct | 64.37% | 87.56% | 78.68% | 29.38% | 72.22% | 77.41% |
| Qwen3-235B-A22B | 67.94% | 87.90% | 77.03% | 40.12% | **83.33%** | 76.32% |
| Qwen3-32B | 69.25% | **88.90%** | 77.83% | 43.12% | 72.22% | 75.79% |
| o3-Mini | 64.61% | 86.15% | 79.08% | 28.75% | 72.22% | 82.96% |
| GPT-4.1 | 68.69% | 85.42% | **79.92%** | 40.50% | 77.78% | **85.95%** |
| GPT-4.5-Preview | 70.32% | 86.12% | 79.34% | 45.38% | 66.67% | 83.64% |
| Qwen2.5-7B-Instruct | 55.10% | 84.19% | 72.32% | 12.88% | 72.22% | 67.93% |
| *with* TOUCAN | 58.26%$^{+3.16\%}$ | 78.52% | 74.50% | 22.62% | 66.67% | 75.18% |
| Qwen2.5-14B-Instruct | 57.69% | 83.38% | 73.70% | 19.75% | **83.33%** | 68.46% |
| *with* TOUCAN | 65.09%$^{+7.40\%}$ | 85.42% | 76.01% | 35.25% | 72.22% | 75.96% |
| Qwen2.5-32B-Instruct | 61.73% | 85.58% | 76.01% | 26.38% | 72.22% | 72.68% |
| *with* TOUCAN | **70.45%**$^{+8.72\%}$ | 87.12% | 78.90% | **46.50%** | 77.78% | 78.10% |
| Llama-3.1-8B-Instruct | 26.23% | 47.96% | 33.63% | 6.38% | 94.44% | 5.26% |
| *with* TOUCAN | 58.46%$^{+32.23\%}$ | 83.44% | 70.68% | 24.88% | 77.78% | 64.85% |
| Llama-3.3-70B-Instruct | 53.03% | 85.23% | 62.86% | 16.38% | 100.00% | 48.50% |
| *with* TOUCAN | 66.20%$^{+13.17\%}$ | 85.79% | 73.48% | 42.25% | 77.78% | 68.22% |

Table 3: This table presents $\tau$-Bench and $\tau^2$-Bench results for models fine-tuned on TOUCAN compared to their respective baselines. Improvements are observed across most evaluation scenarios.

| Model | $\tau$-bench | | | $\tau^2$-bench | | | |
| --- | --- | --- | --- | --- | --- | --- | --- |
| | *Avg.* | *Airline* | *Retail* | *Avg.* | *Airline* | *Retail* | *Telecom* |
| Qwen2.5-7B-Instruct | 15.03% | 8.75% | 21.30% | 16.08% | 14.00% | 17.54% | 16.70% |
| *with* TOUCAN | 22.48%$^{+7.45\%}$ | 15.50% | 29.46% | 17.77%$^{+1.69\%}$ | 20.00% | 22.80% | 10.50% |
| Qwen2.5-14B-Instruct | 30.85% | 17.25% | 44.46% | 24.46% | 12.00% | 41.20% | 20.18% |
| *with* TOUCAN | 35.24%$^{+4.39\%}$ | 22.00% | 48.48% | 30.43%$^{+5.97\%}$ | 22.00% | 49.10% | 20.18% |
| Qwen2.5-32B-Instruct | 38.76% | 26.00% | 51.52% | 29.40% | 18.00% | 49.10% | 21.11% |
| *with* TOUCAN | 42.33%$^{+3.57\%}$ | 29.00% | 55.65% | 31.60%$^{+2.20\%}$ | 22.00% | 52.60% | 20.20% |

# 4 EXPERIMENTS

In this section, we demonstrate the performance of TOUCAN by performing supervised fine-tuning (SFT) on baseline models of different sizes. We then compare the fine-tuned models' performance against existing model baselines across several widely used agentic tool-call benchmarks.

## 4.1 EXPERIMENT SETUP

**Model and Baseline Setup.** We perform supervised fine-tuning on `Qwen2.5-7B-Instruct`, `Qwen2.5-14B-Instruct`, and `Qwen2.5-32B-Instruct` (Team, 2024) to demonstrate the efficacy of TOUCAN across models of varying sizes. Detailed fine-tuning parameters are provided in Appendix C.2. We benchmark the performance of our fine-tuned models against models of comparable or larger scales, including `DeepSeek-V3` DeepSeek-AI et al. (2025), `Qwen2.5-72B-Instruct`, `Qwen3-235B-A22B`, `Qwen3-32B` Yang et al. (2025), and closed-source OpenAI models such as o3-mini, GPT-4.1, and GPT-4.5-Preview.

**TOUCAN Setup.** Given the large volume of the full dataset, we adopted a strategy similar to Xu et al. (2025b) by sampling from a high-quality subset of TOUCAN. This subset was selected based on the following criteria: question quality and scenario realism scores of 5, response completeness and

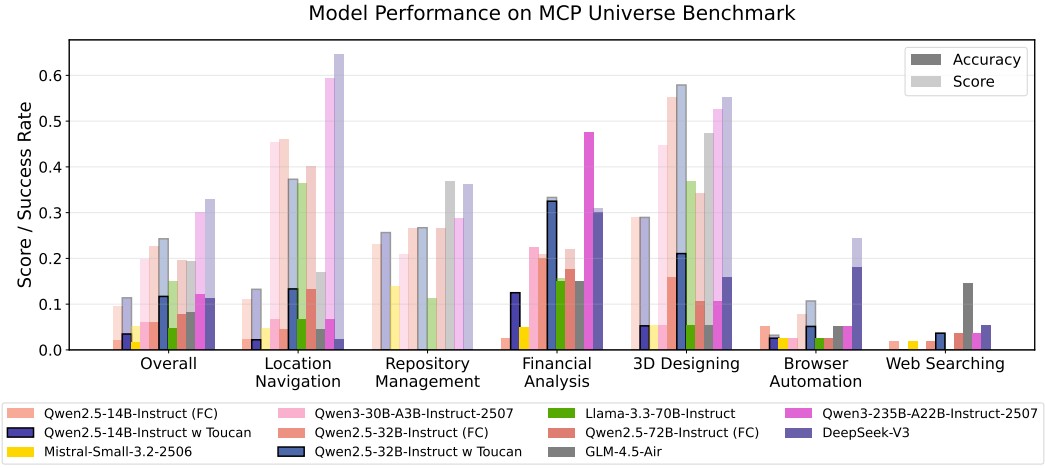

Figure 7: This figure compares the performance of TOUCAN-tuned models with other open-source models on MCP-Universe (Luo et al., 2025). Model sizes increase from left to right. Bars with darker colors represent task success rate (full task completion), while lighter colors represent average evaluation scores considering partial task completion. TOUCAN-tuned models are shown with black borders. TOUCAN-tuned models outperform other models of similar sizes across most tasks.

conciseness scores of at least 4, and desired tool use percentage of 1.0 (indicating that trajectories fully utilize all required tools from the task). We performed necessary data re-balancing to ensure the dataset remains representative across different categories. The resulting SFT dataset comprises 28.3K instances from the original pipeline, 40K instances from Ext.1 (Irrelevance), 15.8K instances from Ext.2 (Diversify), and 35.2K instances from Ext.3 (Multi-Turn), totaling 119.3K instances.

**Benchmarks.** We assess the performance of TOUCAN across several key tool-agentic benchmarks, including BFCL V3 Patil et al. (2025), $\tau$-Bench Yao et al. (2024), $\tau^2$-Bench (Barres et al., 2025), and MCP-Universe Luo et al. (2025). All evaluations are conducted on an $8 \times$ H100 server. For BFCL-V3, we use the official evaluation setup. For $\tau$-Bench and $\tau^2$-Bench, we employ `GPT-4o` as user simulators. For MCP-Universe, we configure the local evaluation environment as specified in the benchmark documentation.

## 4.2 EXPERIMENTAL RESULTS

**TOUCAN Effectively Increases Agentic Tool-Calling Performance.** Tables 2 and 3 present the experimental results of models fine-tuned on TOUCAN across BFCL V3, $\tau$-Bench, and $\tau^2$-Bench, respectively. We make the following key observations: First, models fine-tuned with TOUCAN show performance improvements compared to baseline models without fine-tuning across almost all aspects of these three benchmarks, indicating that TOUCAN effectively enhances the agentic and tool-calling capabilities of models. Second, on BFCL V3, models fine-tuned on TOUCAN outperform larger production LLMs including `DeepSeek-V3` and `GPT-4.5-Preview` in average scores and achieve top performance in the *multi-turn* subset. This demonstrates the effectiveness of TOUCAN and validates our dataset design.

**TOUCAN Enhances Models' Performance on Using Real-World MCP Servers.** Figure 7 demonstrates a performance comparison between TOUCAN-tuned mod-

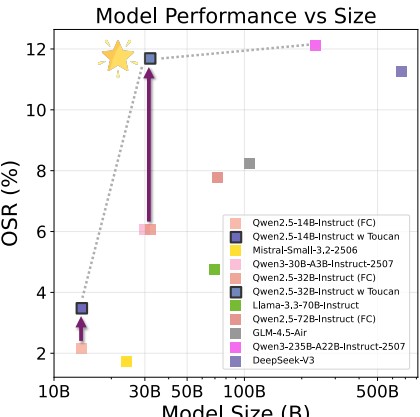

Figure 8: Model Performance vs Size on MCP-Universe Benchmark. We report overall task success rate (OSR). Our models achieve a new Pareto optimum.

els and other open-source models of similar or larger sizes across six domains: Location Navigation, Repository Management, Financial Analysis, 3D Design, Browser Automation, and Web Search. We note that most servers in the benchmark require careful configurations and thus were not included in our data synthesis pipeline. Nevertheless, TOUCAN-tuned models show significant improvements on these challenging tasks compared to baselines, indicating that exposure to diverse tools enhances model performance on agentic tasks. Notably, our 32B model achieves the highest scores in 3D Design and strong performance in Financial Analysis, even outperforming much larger frontier open models like `Llama-3.3-70B-Instruct`, `Qwen2.5-72B-Instruct`, `GLM-4.5-Air` (106B), and `DeepSeek-V3` (671B).

Figure 8 plots model performance versus model size on MCP-Universe benchmark. We observe that TOUCAN -tuned models establish a new Pareto optimum, indicating that TOUCAN can help models achieve superior performance-efficiency trade-offs in agentic tasks.

### 4.3 ABLATION ANALYSIS

To validate our extension designs, we perform ablation analysis on the Qwen2.5-14B-Instruct model, where we fine-tune on progressively extended versions of TOUCAN, allowing us to isolate the contributions of each extension described in Section 3.2. The experimental results are shown in Figrue 9. We observe that all components contribute to improved scores. Detailed benchmark scores for the BFCL ablation study are provided in Appendix C.3. In addition, we include further ablations on tool scaling, dataset scaling, fine-tuning comparisons between TOUCAN and other baseline datasets, as well as trajectory annotation and filtering, in Appendix C.4–C.8.

Figure 9: This table shows ablation analysis of TOUCAN extensions.

| | BFCLv3 | $\tau$-bench | |
| --- | --- | --- | --- |
| | | Airline @1 | Retail @1 |
| Qwen2.5-14B-Instruct | 57.69% | 17.25% | 44.46% |
| + Single Turn | 60.16% | 15.50% | 36.95% |
| + Irrelevance | 64.74% | 16.75% | 41.63% |
| + Diversify | 64.56% | 17.25% | 43.70% |
| + Multi-Turn | 65.09% | 22.00% | 48.48% |

## 5 CONCLUSION AND FUTURE WORK

This paper introduces TOUCAN , a tool-agentic dataset containing 1.5M trajectories designed to train better agentic models. We propose a comprehensive pipeline for data generation and demonstrate that models fine-tuned on TOUCAN achieve superior performance on benchmarks including BFCL-V3 and MCP-Universe. TOUCAN represents the first step in a long-term effort to leverage tool use for building stronger LLM agents. Despite being a valuable contribution, we acknowledge our work exhibits certain limitations, which we plan to address through different initiatives.

**Expanding to More MCP Servers.** While our dataset is comprehensive, it was collected in June 2025, and new servers continue to emerge. We excluded MCP servers requiring special configurations (e.g., requires API keys or account setups), which simplifies the onboarding procedure but may overlook important servers and widely-used scenarios (e.g., Notion and GitHub). Manually onboarding more servers or developing automated onboarding agents could be valuable future work.

**Expert models to simulate tool-responses.** While real tool execution produces higher-quality results, it is often slow and costly, and therefore, not an option for everyone. To provide an alternative that also yields quality, we plan to develop an expert LLM capable of simulating tool execution. This artificial component will significantly reduce the cost of generating trajectory data involving tool use. Although the idea of tool-execution simulation is known within the community, it has most likely been implemented using off-the-shelf, closed-source LLMs.

**MCP Benchmark for web search.** As tool-use capabilities become central to both LLMs and LLM-agents, specific scenarios such as web search have gained prominence in the community as a means of synthesizing complex reasoning tasks. To advance this direction, we plan to develop an MCP benchmark focused on web search capabilities.

## 6 USE OF LARGE LANGUAGE MODELS (LLMs)

In our work, we used large language models (LLMs) to assist with improving the grammar, clarity, and overall readability of the manuscript, as well as to help generate the pipeline diagram included in the paper. All LLM-generated content was thoroughly verified by the authors as part of an iterative process to ensure accuracy, quality, and consistency with the scientific contributions of the work.

## 7 ETHICS STATEMENT

Developers planning to use `Toucan` for LLM fine-tuning should take into account certain considerations.

**Data Ownership and Licensing.** The MCP server specification files used to build TOUCAN were collected in June 2025 from `https://smithery.ai/`, a public platform hosting such specifications. These files were voluntarily published by their owners in accordance with the platform's privacy notice. Given the case a legitimate owner requests removal of their content from our dataset, we will honor that request through a take down process available via our GitHub repository.

**Sensitive Information.** The risk of exposing sensitive data in specification files is minimal, as they generally rely on placeholders rather than real information. However, human error may still lead to the inclusion of URLs, tokens, or email addresses. To mitigate this, we apply a pre-filtering stage with rule-based verifiers that detect common patterns of personally identifiable information (PII).

**Data Evolution.** Our data were collected in June 2025, so TOUCAN captures real-world tool-use scenarios available at that time. For example, responses from search MCP servers reflect information current through June 2025. To facilitate future updates and customization, we provide our modular data pipeline, allowing researchers and practitioners to expand domain coverage and tailor tool representations for their applications.

**LLM Hallucinations.** Only tasks and annotations in TOUCAN were generated with LLMs; trajectories were produced using LLMs in combination with agent frameworks and remote MCP servers. This integration ensures reliable tool call executions and responses, reducing the likelihood of code errors from hallucinations. Nevertheless, hallucinations remain a general risk when using LLMs, and outputs from models fine-tuned with TOUCAN should always be verified by humans.

## 8 REPRODUCIBILITY STATEMENT

We provide the code for our data generation pipeline, along with detailed instructions for executing the pipeline end-to-end, as well as sample dataset files in the supplementary materials. The main paper and appendix further document key implementation details, including prompt templates, hyperparameter configurations used during fine-tuning, extensions of our data analysis and fine-tuning experiments, as well as compute requirements. After publication, we plan to release the full codebase in a public GitHub repository and make our datasets publicly available on the HuggingFace platform.

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

## A  DATASET SCHEMA AND EXAMPLES

An instance of TOUCAN contains the following columns:

- **uuid:** Unique sample identifier.

- **subset:** Annotation specifying which pipeline was used to generate the trajectory. Options: (1) *single-turn-original:* only the core processing (Stage 1 to 5) described in Section 3 are applied, (2) *irrelevant:* a server shuffle process applied on top of the *single-turn-original* pipeline, (3) *single-turn-diversify:* a question diversification process applied on top of the *single-turn-original* pipeline, and (4) *multi-turn:* a multi-turn extension of the *single-turn-original* and *single-turn-diversify* subsets.

- **messages:** The trajectory formatted with the chat template from the original LLM-agent used for generation. The system prompt includes the associated list of tools.

- **question:** The user task crafted to generate the trajectory.

- **target_tools:** The MCP tools used as seeds for question generation.

- **question_quality_assessment:** Task evaluation by an LLM-as-judge, covering quality, difficulty, realism, and uniqueness.

- **response_quality_assessment:** Response evaluation by an LLM-as-judge, covering completeness and conciseness.

- **message_num_rounds:** Total number of messages, including turns of all types.

- **metadata:** Original MCP server data collected and used as seed for generation, as well as respective LLM annotations.

This is the structure of an instance in TOUCAN :

```
{
  "uuid": "3ac8fdcc-b9b5-50d2-a840-947a42b558d2",
  "subset": "single-turn-original",
  "messages": "[{...long JSON string of messages...}]",
  "question": "I'm reviewing my company's policy document that needs
  ↪  quarterly updates for compliance purposes. I need to first
  ↪  understand its current structure with all sections and
  ↪  subsections, then update every occurrence of 'Q3 2024' to 'Q1
  ↪  2025' throughout the document, and finally add explanatory
  ↪  footnotes to the last 3 sections that reference our new
  ↪  regulatory guidance document published last month. Can you help
  ↪  me accomplish these updates systematically?",
  "target_tools": "get_document_outline, search_and_replace,
  ↪  add_footnote_to_document",
  "question_quality_assessment_kimik2w4a16": {
    "tool_selection_difficulty": { "reasoning": "...", "score": 3 },
    "tool_selection_uniqueness": { "reasoning": "...", "score": 3 },
    "question_quality": { "reasoning": "...", "score": 4 },
    "scenario_realism": { "reasoning": "...", "score": 5 },
    "verifiable": { "reasoning": "...", "score": 4 },
    "stability": { "reasoning": "...", "score": 5 },
    "overall_score": 4.0
  },
  "response_quality_assessment_oss120b": {
    "completeness": { "reasoning": "...", "score": 2 },
    "conciseness": { "reasoning": "...", "score": 3 },
    "overall_score": 2.5,
    "desired_tools_used_percentage": 0.3333,
    "order_correctness": false
  },
  "messages_num_rounds": 9,
  "metadata": "{...long metadata JSON string...}"
}
```

Below, we present a TOUCAN trajectory, chat template tokens from the model generator and emojis have been removed, and some sections (e.g., tool list, tool output, assistant response) have been abbreviated to improve readability.

```
[
  {
    "role": "system",
    "content": "{... contains the list of tools ... }"
  },
  {
    "role": "user",
    "content": "Can you explain what ENS (Ethereum Name Service)
    ↪  represents in the context of a transaction and why a name like
    ↪  'blockscout.eth' might appear instead of a raw address?"
  },
  {
    "role": "assistant",
    "content": "I'll explain what ENS (Ethereum Name Service) represents,
    ↪  and show you how it resolves a name like 'blockscout.eth' into
    ↪  its actual Ethereum address."
  },
  {
    "role": "assistant",
    "content": "",
    "function_call": {
      "name": "blockscout-mcp-server-__unlock_blockchain_analysis__",
      "arguments": "{}"
    }
  },
  {
    "role": "function",
    "content": "{... long tool output ...}"
  },
  {
    "role": "assistant",
    "content": "Now, let me resolve 'blockscout.eth' to show the actual
    ↪  address it represents."
  },
  {
    "role": "assistant",
    "content": "",
    "function_call": {
      "name": "blockscout-mcp-server-get_address_by_ens_name",
      "arguments": "{\"name\": \"blockscout.eth\"}"
    }
  },
  {
    "role": "function",
    "name": "blockscout-mcp-server-get_address_by_ens_name",
    "content": "{\n  \"data\": {\n    \"resolved_address\":
    ↪  \"0x441e1e47a6Fa2DBfD3cd9b54291E9AB3a58D7975\"\n  },\n
    ↪  \"data_description\": null,\n  \"notes\": null,\n
    ↪  \"instructions\": null,\n  \"pagination\": null\n}"
  },
  {
    "role": "assistant",
    "content": "{... parsed assistant response ...}"
  },
  {
    "role": "user",
    "content": "Based on that explanation, could you look up the ENS name
    ↪  'blockscout.eth' and provide the Ethereum address it resolves
    ↪  to?"
  },
  {
    "role": "assistant",
```

```
        "content": "I already resolved that for you in my previous response!
    ↪    When I called the ENS lookup tool, `\"blockscout.eth\"` resolved
    ↪    to the Ethereum
    ↪    address:\n\n`0x441e1e47a6Fa2DBfD3cd9b54291E9AB3a58D7975`"
      }
    ]
```

In the previous trajectory example, we abbreviated the assistant's parsed response for readability. The full response is shown below.

```
1  ## What ENS (Ethereum Name Service) Represents in Transactions
2
3  ENS stands for **Ethereum Name Service**, and it functions much like "DNS
    ↪  for Ethereum addresses." It allows users to register human-readable
    ↪  names (like "blockscout.eth") that map to complex hexadecimal
    ↪  blockchain addresses.
4
5  ### Key Points About ENS:
6
7  1. **Human-Readable Names for Complex Addresses**
8     - Instead of displaying `0x441e1e47a6Fa2DBfD3cd9b54291E9AB3a58D7975`
    ↪     (which is the raw address for blockscout.eth), interfaces show the
    ↪     friendly "blockscout.eth"
9     - This reduces errors from copy/pasting long hex addresses
10
11 2. **Reverse Resolution**
12    - When you see "blockscout.eth" in transaction data, wallet labels, or
    ↪     block explorers, it means:
13      - ENS has been configured with this name
14      - The system has performed a reverse lookup to find the name
    ↪       associated with the address
15
16 3. **Self-Sovereign Identity**
17    - ENS names are owned and controlled by users through smart contracts
18    - They can be transferred, updated, or associated with additional
    ↪     metadata (social profiles, avatars, etc.)
19
20 ### Why "blockscout.eth" Appears Instead of Raw Address:
21
22 - **User Experience**: "blockscout.eth" is immediately recognizable and
    ↪   memorable
23 - **Trust Verification**: Users can verify they're interacting with the
    ↪   legitimate Blockscout entity rather than a random address
24 - **Branding**: Organizations use ENS names for consistent branding
    ↪   across DeFi protocols and wallet interfaces
25 - **Error Prevention**: Reduces costly mistakes when users verify
    ↪   transaction recipients
26
27 Every time you see an ENS name like "blockscout.eth" in a transaction,
    ↪   the underlying system has actually resolved this to the precise
    ↪   blockchain address `0x441e1e47a6Fa2DBfD3cd9b54291E9AB3a58D7975`-the
    ↪   ENS simply provides a human-friendly layer on top of the raw
    ↪   cryptographic identity.
```

Below example demonstrated an error recovery of TOUCAN. In this example, the assistant attempts to invoke a math evaluation tool multiple times with incorrect function syntax ($exp$, $e$), receives execution errors from the tool, and then recovers by reformulating the expression using a numeric approximation of $e$ to successfully complete the calculation.

```
[
  {
    "role": "system",
    "content": "{... contains the list of tools ... }"
  },
  {
```

```
        "role": "user",
        "content": "Calculate the company's revenue in year\u202f5 using the
        ↪   exponential model Revenue\u202f=\u202f10,000\u202f\u00d7\u202fe^⌋
        ↪   (0.15\u202f\u00d7\u202fyear). Show your work and give the
        ↪   numerical result."
    },
    {
        "role": "assistant",
        "content": "I'll calculate the company's revenue in year 5 using the
        ↪   exponential growth model you provided. Let me evaluate this
        ↪   expression."
    },
    {
        "role": "assistant",
        "content": "",
        "function_call": {
            "name": "expression-evaluation-server-eval",
            "arguments": "{\"expression\": \"10000 * exp(0.15 * 5)\"}"
        }
    },
    {
        "role": "function",
        "content": "handler returned an error: unknown name exp (1:9)\n |
        ↪   10000 * exp(0.15 * 5)\n | ........^",
        "name": "expression-evaluation-server-eval"
    },
    {
        "role": "assistant",
        "content": "Let me try with the proper function name:"
    },
    {
        "role": "assistant",
        "content": "",
        "function_call": {
            "name": "expression-evaluation-server-eval",
            "arguments": "{\"expression\": \"10000 * e^(0.15 * 5)\"}"
        }
    },
    {
        "role": "function",
        "content": "handler returned an error: unknown name e (1:9)\n | 10000
        ↪   * e^(0.15 * 5)\n | ........^",
        "name": "expression-evaluation-server-eval"
    },
    {
        "role": "assistant",
        "content": "Let me calculate this step by step, using the fact that e
        ↪   \u2248 2.718281828:"
    },
    {
        "role": "assistant",
        "content": "",
        "function_call": {
            "name": "expression-evaluation-server-eval",
            "arguments": "{\"expression\": \"10000 * 2.718281828^(0.15 *
            ↪   5)\"}"
        }
    },
    {
        "role": "function",
        "content": "10000 * 2.718281828^(0.15 * 5) = 21170.00016344546",
        "name": "expression-evaluation-server-eval"
    },
    {
        "role": "assistant",
```

```
        "content": "## Solution\n\n**Given Model:** Revenue = 10,000 \u00d7
    ↪   e^(0.15 \u00d7 year)\n\n**For Year 5:**\n\n1. **Calculate the
    ↪   exponent:** 0.15 \u00d7 5 = 0.75\n\n2. **Calculate e^0.75:**
    ↪   e^0.75 \u2248 2.117\n\n3. **Calculate final revenue:** 10,000
    ↪   \u00d7 2.117 = **$21,170.00**\n\n**Numerical Result:** The
    ↪   company's revenue in year 5 is **$21,170** (rounded to the
    ↪   nearest dollar).\n\nNote: This represents approximately 111.7%
    ↪   growth over the starting revenue of $10,000, demonstrating the
    ↪   power of exponential growth at 15% per year compounded
    ↪   continuously."
},
]
```

## B  MORE ON DATASET ANALYSIS

### B.1  FEATURED SERVERS IN TOUCAN

Table 4 demonstrates featured servers of TOUCAN. We note that these featured servers span a wide range of domains, making combinations possible for complex tool use tasks.

| Server Name | Repository Name | Domain | Tools Count |
|---|---|---|---|
| OKX Server | esshka/okx-mcp | Cryptocurrency & Blockchain | 2 |
| AI Research Assistant - Semantic Scholar | Access via Smithery [2] | Web Search & Research | 10 |
| Book Search Server | Access via Smithery [3] | Web Search & Research | 1 |
| PubMed MCP Server | JackKuo666/PubMed-MCP-Server | Web Search & Research | 4 |
| Flux ImageGen Server | falahgs/flux-imagegen-mcp-server | AI/ML Tools | 3 |
| PokÃ©mcp | NaveenBandarage/poke-mcp | Data Analysis & Processing | 4 |
| Hotel Booking Server | jinkoso/jinko-mcp | E-commerce | 6 |
| Cloudflare Playwright | cloudflare/playwright-mcp | Browser Automation | 24 |
| Time MCP Server | yokingma/time-mcp | Time & Calendar | 6 |
| Exa Search | exa-labs/exa-mcp-server | Web Search & Research | 8 |
| Weather Forecast Server | iremaltunay55/deneme | Weather | 5 |
| Advanced Calculator Server | alan5543/calculator-mcp | Data Analysis & Processing | 17 |
| Dictionary Server | ceydasimsekk/dictionarymcp | Others | 1 |
| Airbnb Search and Listing Details Server | AkekaratP/mcp-server-airbnb | Web Search & Research | 2 |
| Code Runner MCP Server | formulahendry/mcp-server-code-runner | Development Tools | 1 |
| Movie Recommender | iremert/movie-recommender-mcp | Content Creation | 1 |
| United States Weather | smithery-ai/mcp-servers | Weather | 6 |
| Context7 | upstash/context7-mcp | Development Tools | 2 |
| Think Tool Server | PhillipRt/think-mcp-server | Memory Management | 1 |
| OpenAPI MCP Server | janwilmake/openapi-mcp-server | API Integration | 2 |
| Film Information Server | zehranurugurr/film_mcp | Content Creation | 1 |
| Trends Hub | baranwang/mcp-trends-hub | News & Media | 21 |
| ClinicalTrials MCP Server | JackKuo666/ClinicalTrials-MCP-Server | Health & Fitness | 7 |
| Drawing Tool for AI Assistants | flrngel/mcp-painter | Content Creation | 4 |
| LeetCode | jinzcdev/leetcode-mcp-server | Development Tools | 9 |

Table 4: Featured Server Information

### B.2  MORE ON MCP SERVER ANALYSIS IN TOUCAN

Figure 10 shows the distribution of the most frequently used MCP servers in our dataset, highlighting the diversity of servers and domains covered in TOUCAN. Figure 11 shows the distribution of tool counts across the 495 MCP servers employed by TOUCAN , revealing that most servers expose only a limited number of tools, with the majority containing fewer than 10 tools.

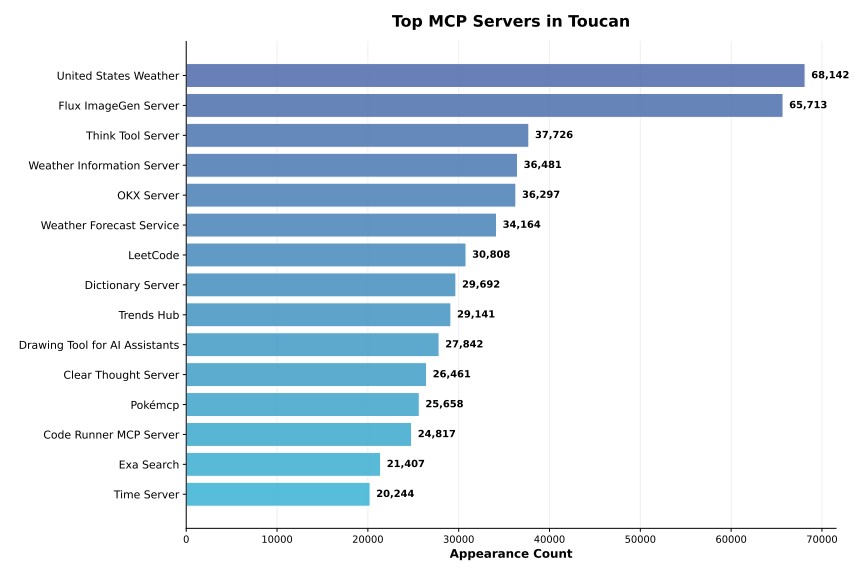

Figure 10: Distribution of the most frequently occurring MCP servers in the TOUCAN dataset.

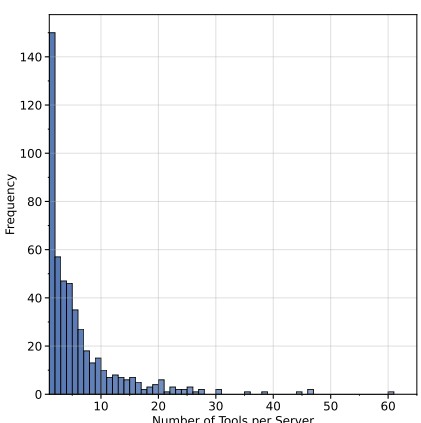

Figure 11: Tools Number distribution across MCP servers

## B.3 EMEDDING VISUALIZATION

Figure 12 presents embedding visualization via Embedding Atlas (Ren et al., 2025) using the `Xenova/multilingual-e5-small` embedding model with UMAP projection McInnes & Healy (2018). The visualization demonstrates that TOUCAN covers a wide range of topics. In addition, the proposed TOUCAN extensions (e.g., diversification) effectively increase the overall dataset coverage.

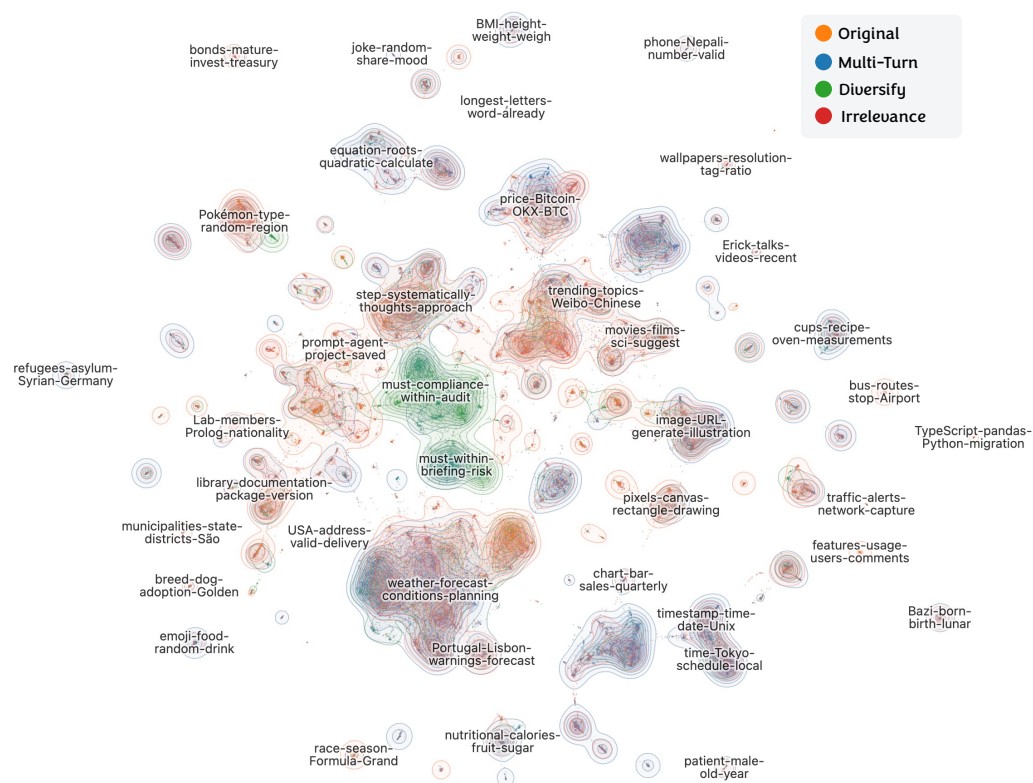

Figure 12: This figure is the visualization of 50K random-sampled TOUCAN instances via Embedding Atlas (Ren et al., 2025).

### B.4 DOMAIN COVERAGE COMPARISON BETWEEN TOUCAN AND MCP UNIVERSE

In what follows, we compare the MCP server names included in TOUCAN with those used for constructing MCP Universe. The results are summarized in Table 5. Our analysis shows that four MCP Universe domains are completely out-of-distribution (OOD) with respect to TOUCAN, indicating that our fine-tuned models demonstrate strong generalization performance on domains that were never seen during training.

Table 5: In-distribution (ID) and out-of-distribution (OOD) domain coverage of TOUCAN relative to MCP Universe.

| Benchmark | Benchmark Domain | TOUCAN |
|---|---|---|
| MCP Universe | Location Navigation | OOD |
| | Repository Management | OOD |
| | Financial Analysis | ID |
| | 3D Design | OOD |
| | Browser Automation | ID |
| | Web Searching | OOD |

## C MORE ON EXPERIMENTS

### C.1 LLM ANNOTATION

Figure 13 shows the Pearson correlation between human annotations and LLM-as-a-judge evaluations across different models on 50 randomly sampled instances. We observe that GPT-4.1 and Kimi-K2 achieve the highest correlation with human judgments. Notably, the Pearson correlation between the two human annotators is only 0.5028, indicating moderate inter-annotator agreement, especially on inherently subjective aspects such as tool selection uniqueness and scenario realism. This suggests that the relatively low model-human correlation is partly due to annotation subjectivity rather than model unreliability. Considering cost efficiency and maintaining an end-to-end open-source pipeline, we deploy Kimi-K2 locally as the annotator. Our annotation prompt is available in Appendix D.4.

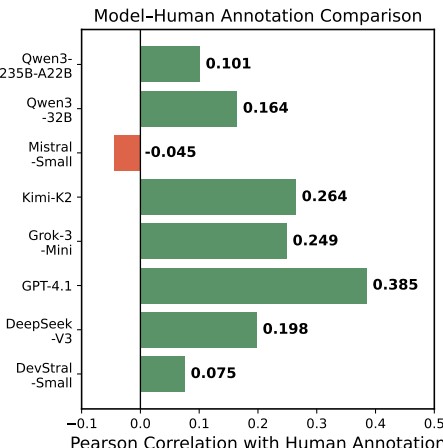

Figure 13: Pearson correlation between human annotator and LLM-as-a-Judge evaluations across different models.

### C.2 FINE-TUNING HYPER-PARAMETERS

We fine-tune models with TOUCAN using a super computing cluster, which is outfitted with NVIDIA H100 GPUs. The fine-tuning hyper-parameters can be found in Table 6.

Table 6: This table shows the hyper-parameters for supervised fine-tuning.

| Hyper-parameter | Value |
|---|---|
| Tool-Call Template | Hermes |
| Learning Rate | $2 \times 10^{-5}$ |
| Number of Epochs | 2 |
| Number of Devices | 8 or 64 |
| Per-device Batch Size | 1 |
| Gradient Accumulation Steps | 8 (8 GPUs) or 1 (64 GPUs) |
| Effective Batch Size | 64 |
| Optimizer | Adamw with $\beta s = (0.9, 0.999)$ and $\epsilon = 10^{-8}$ |
| Deepspeed | zero3 |
| Max Sequence Length | 32768 |

## C.3 ABLATION STUDIES ON DATA EXTENSIONS

Table 7 details the individual scores of the BFCL V3 benchmark for our ablation analysis. We observe that all extensions are meaningful in improving model performance.

Table 7: Ablation of TOUCAN Extensions on BFCL V3 Benchmark.

| | Overall | Single Turn | | Multi Turn | Hallucination | |
| | | *Non-live (AST)* | *Live (AST)* | | *Relevance* | *Irrelevance* |
|---|---|---|---|---|---|---|
| Qwen2.5-14B-Instruct | 57.69% | 83.38% | 73.70% | 19.75% | 83.33% | 68.46% |
| + Single Turn | 60.16% | 87.50% | 66.86% | 34.38% | 72.22% | 46.88% |
| + Irrelevance | 64.74% | 88.46% | 77.25% | 30.38% | 72.22% | 77.85% |
| + Diversify | 64.56% | 86.06% | 76.90% | 32.50% | 72.22% | 75.45% |
| + Multi-Turn | 65.09% | 85.42% | 76.01% | 35.25% | 72.22% | 75.96% |

## C.4 ABLATION STUDIES ON TOOL SCALING

In this experiment, we create five subsets of TOUCAN datasets, each doubling the number of tools relative to the previous one. The number of tools preserved in the dataset ranges from 100 to 1,600, while the total number of trajectories is kept constant at 20,000. We fine tuned `Qwen2.5-32B-Instruct` on these subsets, and evaluate the resulting models on BFCL V3 and $\tau$-Bench. Table 8 shows the results of this experiment. We observe a consistent upward trend in overall performance as tool diversity increases, indicating that a larger and more diverse tool set leads to better generalization rather than redundant learning.

Table 8: Ablation on Tool Diversity on BFCL V3 and $\tau$-Bench.

| | BFCL-V3 | | | | | | $\tau$-Bench | |
| Model Variant | Overall | Single Turn | | Multi Turn | Hallucination | | Airline @1 | Retail @1 |
| | | *Non-live (AST)* | *Live (AST)* | | *Relevance* | *Irrelevance* | | |
|---|---|---|---|---|---|---|---|---|
| Qwen2.5-32B-Instruct-Toucan-100Tools-20K | 60.38% | 87.58% | 64.82% | 36.00% | **88.89%** | 46.66% | 28.50% | 47.39% |
| Qwen2.5-32B-Instruct-Toucan-200Tools-20K | 60.90% | 86.56% | 65.44% | 37.00% | **88.89%** | 49.50% | 27.50% | 50.43% |
| Qwen2.5-32B-Instruct-Toucan-400Tools-20K | 61.99% | 87.48% | 65.08% | 40.00% | 83.33% | 49.00% | 28.50% | 48.48% |
| Qwen2.5-32B-Instruct-Toucan-800Tools-20K | 61.73% | **87.31%** | 64.73% | **40.38%** | 83.33% | 45.98% | 29.25% | 50.54% |
| Qwen2.5-32B-Instruct-Toucan-1600Tools-20K | **62.26%** | 86.27% | **67.57%** | 39.38% | 83.33% | **52.08%** | **29.75%** | **52.06%** |

## C.5 ABLATION STUDIES ON LARGE DATASETS

We conduct an ablation experiment to investigate the relevance of TOUCAN 's large size for the research community. Specifically, we created two training datasets: TOUCAN -Full (including 1.5M trajectories), and TOUCAN -SFT (as detailed in Section 4.1). We fine-tuned `Qwen2.5-14B-Instruct` with each training dataset. Table 9 shows the evaluation results for BFCL V3. Overall, TOUCAN -Full slightly outperforms the SFT subset, and shows a remarkable improvement in the multi-turn setting. Our results also show that the model fine-tuned on the full dataset achieves a lower score on the Irrelevance setting, which suggests that the carefully rebalanced TOUCAN -SFT is more effective at reducing hallucinations.

Table 9: BFCL-V3 Results for TOUCAN Full and SFT datasets.

| Dataset | Overall | Single Turn | | Multi Turn | Hallucination | |
| | | *Non-live (AST)* | *Live (AST)* | | *Relevance* | *Irrelevance* |
|---|---|---|---|---|---|---|
| Qwen2.5-14B-Instruct | 57.69% | 83.38% | 73.70% | 19.75% | 83.33% | 68.46% |
| *with* Toucan-SFT | **65.09%** | **85.42%** | **76.01%** | 35.25% | 72.22% | **75.96%** |
| *with* Toucan-Full | **65.17%** | 84.90% | 74.63% | **39.13%** | **83.33%** | 68.71% |

## C.6 ABLATION STUDIES ON DATA SCALING

We perform a data-scale ablation by randomly sampling subsets of 20K, 40K, 60K, 80K, and 100K trajectories from TOUCAN -SFT (see Section 4.1). We then fine-tune `Qwen2.5-32B-Instruct`

on each subset and compare the results against training on the full dataset. We evaluated the results on the BFCL-V3 benchmark. Table 10 shows the results of this experiment. Overall, the models show a consistent performance gain as the data scale increases, with especially strong improvements in the multi-turn setting. We also observe diminishing returns and near-saturation beyond approximately 80K trajectories. This behavior mirrors common scaling trends in instruction tuning and suggests that our rebalanced subset already provides an effective cost-performance sweet spot.

Table 10: Ablation on SFT Data Scale on BFCL V3 and $\tau$-Bench.

| Model Variant | BFCL-V3 | | | | | | $\tau$-Bench | |
| | Overall | Single Turn | | Multi Turn | Hallucination | | Airline @1 | Retail @1 |
| | | *Non-live (AST)* | *Live (AST)* | | *Relevance* | *Irrelevance* | | |
|---|---|---|---|---|---|---|---|---|
| Qwen2.5-32B-Instruct | 61.73% | 85.58% | 76.01% | 26.38% | 72.22% | 72.68% | 26.00% | 51.52% |
| Toucan-SFT-20K | 68.21% | 88.52% | 74.99% | 42.50% | 83.33% | 74.73% | 27.75% | 47.50% |
| Toucan-SFT-40K | 68.82% | 86.77% | 77.30% | 43.50% | 77.78% | 76.62% | 28.50% | 53.37% |
| Toucan-SFT-60K | 68.55% | 86.71% | 77.08% | 43.12% | 83.33% | 75.87% | 28.25% | 56.30% |
| Toucan-SFT-80K | 69.62% | 87.02% | 77.65% | 45.25% | 77.78% | 77.23% | 30.00% | 58.26% |
| Toucan-SFT-100K | 69.83% | 86.44% | 78.76% | 45.25% | 77.78% | 77.91% | 28.00% | 56.66% |
| Toucan-SFT-119K | **70.45%** | **87.12%** | **78.90%** | **46.50%** | 77.78% | **78.10%** | 29.00% | 55.65% |

## C.7 ABLATION STUDIES WITH COMPARABLE TOOL-CALLING DATASET

We perform a controlled experiment to compare TOUCAN with Nemotron-SFT (tool subset) under a similar data scale (see Table 1) and using the same model, `Qwen2.5-14B-Instruct`, as baseline. While the overall score of Nemotron-SFT achieves a comparable performance to TOUCAN on BFCL V3, our dataset shows substantially stronger performance on other tool-agentic benchmarks, especially $\tau$-Bench and $\tau^2$-bench. These benchmarks better reflect multi-tool reasoning and cross-domain generalization. Table 11 and Table 12 respectively report the results obtained.

Table 11: Comparison between Nemotron-SFT (tool subset) and TOUCAN on BFCL-V3.

| Model | Overall | Single Turn | | Multi Turn | Hallucination | |
| | | *Non-live (AST)* | *Live (AST)* | | *Relevance* | *Irrelevance* |
|---|---|---|---|---|---|---|
| Qwen2.5-14B *with* TOUCAN | 65.09% | 85.42% | 76.01% | 35.25% | 72.22% | 75.96% |
| Qwen2.5-14B *with* Nemotron-SFT(tools) | 65.64% | 85.02% | 81.83% | 30.00% | 66.67% | 85.45% |

Table 12: Comparison between Nemotron-SFT (tool subset) and TOUCAN on $\tau$- and $\tau^2$-Bench.

| Model | $\tau$-bench | | | $\tau^2$-bench | | | |
| | *Avg.* | *Airline* | *Retail* | *Avg.* | *Airline* | *Retail* | *Telecom* |
|---|---|---|---|---|---|---|---|
| Qwen2.5-14B *with* TOUCAN | 35.24% | 22.00% | 48.48% | 30.43% | 22.00% | 49.10% | 20.18% |
| Qwen2.5-14B *with* Nemotron-SFT(tools) | 24.38% | 18.00% | 30.76% | 20.23% | 16.00% | 36.80% | 7.90% |

## C.8 ABLATION ON THE IMPORTANCE OF TRAJECTORY ANNOTATION AND FILTERING

We conducted an ablation study comparing datasets with and without the filtering step (Stage 5) of our generation pipeline. We report results in Table 13. We observed that the overall peformance on BFCL V3, as well as the multi-turn and irrelevance setting benefit from filtering, which confirms the value of including an automated process to filter-out low-quality samples in data generation pipelines.

Table 13: Ablation on Stage 5 on BFCL-V3 Benchmark.

| Model | Overall | Single Turn | | Multi Turn | Hallucination | |
| | | *Non-live (AST)* | *Live (AST)* | | *Relevance* | *Irrelevance* |
|---|---|---|---|---|---|---|
| Qwen2.5-14B-Instruct (FC) | 57.69% | 83.38% | 73.70% | 19.75% | 83.33% | 68.46% |
| TOUCAN *without* Filtering (4 stages) | 62.60% | **86.83%** | 72.01% | 32.25% | 77.78% | 67.01% |
| TOUCAN *with* Filtering (5 stages) | **65.09%** | 85.42% | **76.01%** | **35.25%** | 72.22% | **75.96%** |

# D PROMPTS

## D.1 MCP SERVER ANNOTATION PROMPT

Below is the prompt for annotating MCP server categories.

```
1  ## Task
2  Generate **Server Labels** to categorize the provided MCP Server based on
   ↪  its description and available tools.
3
4  ## Objective
5  Analyze the provided MCP Server's description and available tools, then
   ↪  assign appropriate category labels that best describe its primary
   ↪  functionality and use cases.
6
7  ## Guidelines
8
9  ### Label Selection
10 - Analyze the MCP Server's core functionality and purpose
11 - Consider the types of tools it provides and the problems it solves
12 - Select labels that accurately represent the server's primary use cases
13 - Choose from predefined categories when applicable, but also consider
   ↪  custom labels for unique functionality
14
15 ### Predefined Categories
16 Choose from these established categories when appropriate:
17 - **Web Search & Research**: Tools for searching the web, gathering
   ↪  information, academic research
18 - **Browser Automation**: Web scraping, automated browsing, page
   ↪  interaction
19 - **Memory Management**: Data storage, retrieval, knowledge bases,
   ↪  note-taking
20 - **Operating System**: File operations, system commands, process
   ↪  management
21 - **Data Analysis & Processing**: Analytics, data transformation,
   ↪  statistical analysis
22 - **Cryptocurrency & Blockchain**: Trading, wallet management, DeFi,
   ↪  blockchain interaction
23 - **Daily Productivity**: Task management, scheduling, personal
   ↪  organization
24 - **File Management**: File operations, document handling, storage
   ↪  management
25 - **Database Operations**: Data querying, database management, SQL
   ↪  operations
26 - **API Integration**: Third-party service integration, webhook handling
27 - **Communication Tools**: Messaging, email, notifications, social
   ↪  interaction
28 - **Development Tools**: Code analysis, debugging, version control, CI/CD
29 - **Security & Authentication**: Password management, encryption, access
   ↪  control
30 - **Cloud Services**: Cloud platform integration, serverless functions
31 - **AI/ML Tools**: Machine learning, model interaction, AI-powered
   ↪  features
32 - **Content Creation**: Writing, editing, media generation, publishing
33 - **Social Media**: Social platform integration, posting, analytics
34 - **Financial Services**: Banking, payments, financial data, accounting
35 - **E-commerce**: Shopping, product management, order processing
36 - **Gaming**: Game-related tools, entertainment, interactive features
37 - **Education**: Learning tools, course management, educational content
38 - **Health & Fitness**: Health monitoring, fitness tracking, medical
   ↪  tools
39 - **Travel & Maps**: Location services, travel planning, navigation
40 - **News & Media**: News aggregation, media consumption, journalism tools
41 - **Weather**: Weather data, forecasting, climate information
42 - **Time & Calendar**: Scheduling, time management, calendar integration
```

```
43
44  ### Custom Labels
45  - If the server doesn't fit well into predefined categories, create a
    ↪   custom label
46  - Custom labels should be descriptive and specific to the server's unique
    ↪   functionality
47  - Use clear, concise terminology that would be useful for clustering and
    ↪   organization
48
49  ### Output Requirements
50  - **Primary Label**: The main category that best describes the server
    ↪   (from predefined list or custom)
51  - **Secondary Labels**: Additional relevant categories (0-2 labels)
52  - **Custom Label**: A free-form descriptive label if the server has
    ↪   unique functionality not covered by predefined categories
53
54  ## MCP Server Description
55  {MCP_SERVER_NAME}: {MCP_SERVER_DESCRIPTION}
56
57  Available Tools:
58  {TOOL_LIST}
59
60  ## Output
61  Provide your response in the following XML format:
62
63  <response>
64    <analysis>
65      <!-- Briefly analyze the MCP Server's core functionality and the
         ↪   types of problems it solves based on its description and
         ↪   available tools. -->
66    </analysis>
67    <reasoning>
68      <!-- Brief explanation of why these labels were chosen and how they
         ↪   represent the server's functionality -->
69    </reasoning>
70    <primary_label>
71      <!-- The main category that best describes this server's primary
         ↪   functionality -->
72    </primary_label>
73    <secondary_labels>
74      <!-- Additional relevant categories (0-2 labels), separated by commas
         ↪   if multiple -->
75    </secondary_labels>
76    <custom_label>
77      <!-- A free-form descriptive label if the server has unique
         ↪   functionality not covered by predefined categories. Leave empty
         ↪   if not needed. -->
78    </custom_label>
79  </response>
```

## D.2 TASK GENERATION PROMPT

Below is an example of a task generation prompt for the single-server task synthesis. The prompt generates a question targeting **one tool**.

```
1  ## Task
2  Generate a **Tool Use Question** based on the provided MCP Server and its
   ↪   tool descriptions.
3
4  ## Objective
5  Analyze the provided MCP Server and its available tools, then create a
   ↪   realistic user question that would naturally require the use of one
   ↪   of these tools to solve.
6
7  ## Guidelines
```

```
8
9    ### Question Realism
10   - Create questions that represent real-world scenarios where users would
     ↪   need to interact with the MCP Server's tools
11   - The question should sound natural and authentic, as if asked by someone
     ↪   genuinely needing to accomplish a task
12   - Consider common use cases, problems, or workflows that would require
     ↪   the functionality provided by the MCP Server's tools
13
14   ### Tool Selection
15   - Focus on **ONE specific tool** from the MCP Server that would be most
     ↪   appropriate to answer the question
16   - Choose tools based on the core functionality they provide and how they
     ↪   would solve real user problems
17   - Consider each tool's description and purpose when crafting the question
18
19   ### Question Complexity
20   - Create questions that are clear and specific enough to warrant tool
     ↪   usage
21   - Avoid overly simple questions that could be answered without tools
22   - Include relevant context or constraints that make the tool usage
     ↪   necessary
23   - Do not contain the exact tool name in the question
24
25   ### Output Format
26   Your response should include:
27   1. **Tool Analysis**: Briefly analyze the MCP Server's available tools
     ↪   and their main functionalities.
28   2. **Target Tool**: The specific tool name from the MCP Server that
     ↪   should be used to answer this question.
29   3. **Question**: A clear, realistic user question that requires tool
     ↪   usage.
30
31   ## MCP Server Description
32   {MCP_SERVER_NAME}: {MCP_SERVER_DESCRIPTION}
33
34   Available Tools:
35   {TOOL_LIST}
36
37   ## Output
38   Provide your response in the following XML format:
39
40   <response>
41     <server_analysis>
42       <!-- Briefly analyze the MCP Server's available tools and their main
        ↪   functionalities. -->
43     </server_analysis>
44     <target_tool>
45       <!-- The specific tool name from the MCP Server that should be used
        ↪   to answer this question. -->
46     </target_tool>
47     <question>
48       <!-- A clear, realistic user question that requires tool usage. -->
49     </question>
50   </response>
```

Below is an example of a task generation prompt for the single-server task synthesis. The prompt generates a question targeting **multiple tools**.

```
1    ## Task
2    Generate a **Tool Use Question** based on the provided MCP Server and its
     ↪   tool descriptions.
3
4    ## Objective
```

```
5   Analyze the provided MCP Server and its available tools, then create a
    ↪   realistic user question that would naturally require the use of
    ↪   **{NUM_TOOLS} tools** from this MCP Server to solve completely.
6
7   ## Guidelines
8
9   ### Question Realism
10  - Create questions that represent real-world scenarios where users would
    ↪   need to interact with the MCP Server's tools
11  - The question should sound natural and authentic, as if asked by someone
    ↪   genuinely needing to accomplish a task
12  - Consider common use cases, problems, or workflows that would require
    ↪   the functionality provided by the MCP Server's tools
13
14  ### Tool Selection
15  - Focus on **{NUM_TOOLS} tools** from the MCP Server that would work
    ↪   together to answer the question
16  - The question should require a sequence or combination of tool calls to
    ↪   solve completely
17  - Choose tools based on how they complement each other and create a
    ↪   logical workflow
18  - Consider each tool's description and purpose when crafting the question
    ↪   that requires multiple steps
19
20  ### Question Complexity
21  - Create questions that are complex enough to warrant using {NUM_TOOLS}
    ↪   tools
22  - The question should have multiple components or require several steps
    ↪   to solve
23  - Include relevant context or constraints that make the multi-tool usage
    ↪   necessary
24  - Do not contain the exact tool names in the question
25  - Ensure the question cannot be reasonably answered with just a single
    ↪   tool
26
27  ### Output Format
28  Your response should include:
29  1. **Tool Analysis**: Briefly analyze the MCP Server's available tools
    ↪   and their main functionalities.
30  2. **Target Tools**: The specific tool names from the MCP Server that
    ↪   should be used together to answer this question, in the order they
    ↪   would likely be called.
31  3. **Question**: A clear, realistic user question that requires multiple
    ↪   tool usage.
32
33  ## MCP Server Description
34  {MCP_SERVER_NAME}: {MCP_SERVER_DESCRIPTION}
35
36  Available Tools:
37  {TOOL_LIST}
38
39  ## Output
40  Ensure your question requires exactly {NUM_TOOLS} tools to solve
    ↪   completely. Provide your response in the following XML format:
41
42  <response>
43    <server_analysis>
44      <!-- Briefly analyze the MCP Server's available tools and their main
         ↪   functionalities. -->
45    </server_analysis>
46    <target_tools>
47      <!-- The specific tool names from the MCP Server that should be used
         ↪   together to answer this question, listed in order. e.g.,
         ↪   <tool>create_twitter_post</tool> <tool>get_last_tweet</tool> -->
48    </target_tools>
49    <question>
```

```
50      <!-- A clear, realistic user question that requires multiple tool
        ↪   usage. -->
51    </question>
52  </response>
```

Below is an example of a task generation prompt for the multi-server task synthesis.

```
1   ## Task
2   Generate a **Multi-Server Tool Use Question** based on the provided MCP
    ↪   Servers and their tool descriptions.
3
4   ## Objective
5   Analyze the provided MCP Servers and their available tools, then create a
    ↪   realistic user question that would naturally require the use of
    ↪   **{NUM_TOOLS} tools from at least 2 different MCP servers** to solve
    ↪   completely.
6
7   ## Guidelines
8
9   ### Question Realism
10  - Create questions that represent real-world scenarios where users would
    ↪   need to interact with tools from multiple MCP Servers
11  - The question should sound natural and authentic, as if asked by someone
    ↪   genuinely needing to accomplish a complex task
12  - Consider workflows that span across different services/domains that
    ↪   would require multiple servers
13  - Think about how different MCP servers complement each other in
    ↪   real-world use cases
14
15  ### Server and Tool Selection
16  - Use tools from **at least 2 different MCP servers** to answer the
    ↪   question
17  - Select **{NUM_TOOLS} tools total** that work together across multiple
    ↪   servers
18  - The question should require a sequence or combination of tool calls
    ↪   from different servers to solve completely
19  - Choose tools based on how they complement each other across different
    ↪   services/domains
20  - Consider each tool's description and purpose when crafting the
    ↪   cross-server workflow
21  - Ensure tools from different servers create a logical, interconnected
    ↪   workflow
22
23  ### Question Complexity
24  - Create questions that are complex enough to warrant using {NUM_TOOLS}
    ↪   tools across multiple servers
25  - The question should have multiple components or require several steps
    ↪   that span different services
26  - Include relevant context or constraints that make the multi-server tool
    ↪   usage necessary
27  - Do not contain the exact tool names or server names in the question
28  - Ensure the question cannot be reasonably answered with tools from just
    ↪   a single server
29  - Create scenarios that naturally require different types of services
    ↪   working together
30
31  ### Cross-Server Integration
32  - Think about how different servers' capabilities can be combined
33  - Consider data flow between different services (e.g., retrieving data
    ↪   from one service to use in another)
34  - Create realistic scenarios where multiple services need to work
    ↪   together
35  - Focus on complementary functionalities across different domains
36
37  ### Output Format
38  Your response should include:
```

```
1.  **Server Analysis**: Briefly analyze all MCP Servers and their
↪   available tools, focusing on how they can work together.
2.  **Cross-Server Workflow**: Describe the workflow showing how tools
↪   from different servers will be used together.
3.  **Target Tools**: The specific tool names from different MCP Servers
↪   that should be used together, in the order they would likely be
↪   called, with their server names.
4.  **Question**: A clear, realistic user question that requires
↪   multi-server tool usage.

## Available MCP Servers

{SERVER_DESCRIPTIONS}

## Output
Ensure your question requires exactly {NUM_TOOLS} tools from at least 2
↪   different servers to solve completely. Provide your response in the
↪   following XML format:

<response>
  <server_analysis>
    <!-- Briefly analyze all MCP Servers and their available tools,
    ↪   focusing on how they can work together across different
    ↪   domains/services. -->
  </server_analysis>
  <cross_server_workflow>
    <!-- Describe the workflow showing how tools from different servers
    ↪   will be used together to solve the question. -->
  </cross_server_workflow>
  <target_tools>
    <!-- The specific tool names from different MCP Servers that should
    ↪   be used together, listed in order with their server names. e.g.,
    ↪   <tool server="Server1">search_posts</tool> <tool
    ↪   server="Server2">send_email</tool> -->
  </target_tools>
  <question>
    <!-- A clear, realistic user question that requires multi-server tool
    ↪   usage spanning different services/domains. -->
  </question>
</response>
```

Below is an example of a task generation prompt for the task synthesis for featured servers.

```
## Task
Generate a **Multi-Server Tool Use Question** based on featured MCP
↪   Servers and their tool descriptions.

## Objective
Brainstorm a compelling real-world scenario, then analyze the provided
↪   featured MCP Servers and their available tools to create a realistic
↪   user question that would naturally require the use of **{NUM_TOOLS}
↪   tools from at least 2 different MCP servers** to solve completely.

## Guidelines

### Scenario Brainstorming
- Think of realistic, specific scenarios where someone would need to use
↪   {NUM_TOOLS} different tools across multiple servers to accomplish a
↪   meaningful task
- Consider diverse real-world contexts such as:
  - Content creators managing their online presence across different
  ↪   platforms
  - Researchers gathering and analyzing information from multiple sources
  - Developers building and deploying applications using different
  ↪   services
```

```
15    - Business professionals managing projects and communications across
      ↪  platforms
16    - Students working on complex assignments requiring multiple tools
17    - Entrepreneurs launching new ventures using various services
18  - The scenario should be detailed and authentic, representing genuine use
    ↪  cases that span multiple services
19
20  ### Question Realism
21  - Create questions that represent real-world scenarios where users would
    ↪  genuinely need tools from multiple MCP servers
22  - The question should sound natural and authentic, as if asked by someone
    ↪  with a specific goal
23  - Include relevant context, constraints, and details that make the
    ↪  question engaging
24  - Consider workflows that require multiple complementary tools working
    ↪  together across different services
25  - Think about how different servers support each other in real-world use
    ↪  cases
26
27  ### Server and Tool Selection
28  - Use tools from **at least 2 different MCP servers** to answer the
    ↪  question
29  - Select **{NUM_TOOLS} tools total** that work together across multiple
    ↪  servers
30  - The question should require a sequence or combination of tool calls
    ↪  from different servers to solve completely
31  - Choose tools based on how they complement each other across different
    ↪  services/domains
32  - Consider each tool's description and purpose when crafting the
    ↪  cross-server workflow
33  - Ensure tools from different servers create a logical, interconnected
    ↪  workflow
34
35  ### Question Complexity
36  - Create questions that are complex enough to warrant using {NUM_TOOLS}
    ↪  tools across multiple servers
37  - The question should have multiple components or require several steps
    ↪  that span different services
38  - Include relevant context or constraints that make the multi-server tool
    ↪  usage necessary
39  - Do not contain the exact tool names or server names in the question
40  - Ensure the question cannot be reasonably answered with tools from just
    ↪  a single server
41  - Create scenarios that naturally require different types of services
    ↪  working together
42
43  ### Cross-Server Integration
44  - Think about how different servers' capabilities can be combined
45  - Consider data flow between different services (e.g., retrieving data
    ↪  from one service to use in another)
46  - Create realistic scenarios where multiple services need to work
    ↪  together
47  - Focus on complementary functionalities across different domains
48
49  ### Output Format
50  Your response should include:
51  1. **Server Analysis**: Briefly analyze the featured MCP Servers and
    ↪  their available tools, focusing on how they can work together.
52  2. **Cross-Server Workflow**: Describe the workflow showing how tools
    ↪  from different servers will be used together.
53  3. **Target Tools**: The specific tool names from different MCP Servers
    ↪  that should be used together, in the order they would likely be
    ↪  called, with their server names.
54  4. **Question**: A clear, realistic user question that requires
    ↪  multi-server tool usage.
55
```

```
56  ## Available Featured MCP Servers
57
58  {FEATURED_SERVER_DESCRIPTIONS}
59
60  ## Output
61  Ensure your question requires exactly {NUM_TOOLS} tools from at least 2
    ↪  different servers to solve completely. Provide your response in the
    ↪  following XML format:
62
63  <response>
64    <server_analysis>
65      <!-- Briefly analyze the featured MCP Servers and their available
      ↪  tools, focusing on how they can work together across different
      ↪  domains/services. -->
66    </server_analysis>
67    <cross_server_workflow>
68      <!-- Describe the workflow showing how tools from different servers
      ↪  will be used together to solve the question. -->
69    </cross_server_workflow>
70    <target_tools>
71      <!-- The specific tool names from different MCP Servers that should
      ↪  be used together, listed in order with their server names. e.g.,
      ↪  <tool server="Server1">search_posts</tool> <tool
      ↪  server="Server2">send_email</tool> -->
72    </target_tools>
73    <question>
74      <!-- A clear, realistic user question that requires multi-server tool
      ↪  usage spanning different services/domains. -->
75    </question>
76  </response>
```

## D.3 TASK DIVERSIFICATION PROMPT

The following prompt aims to add diversity to the given task by introducing new contexts and personas.

```
1   ## Task
2   Generate **augmented variations** of a given question that maintain the
    ↪  same target tool(s) usage and complexity level but apply them across
    ↪  different contexts and scenarios.
3
4   ## Objective
5   Take an existing question and its associated target tool(s), then create
    ↪  multiple variations that:
6   - Use the same target tool(s) to achieve the core goal
7   - Maintain the exact same tool usage order and final outcome
8   - Apply the question to completely different contexts, scenarios, or
    ↪  domains
9   - Keep the same level of complexity and constraints as the original
10  - Demonstrate how the same tool usage pattern applies across diverse
    ↪  real-world scenarios
11
12  ## Guidelines
13  - Translate the question to distinctly different domains, user personas,
    ↪  or situational contexts while preserving its original complexity
    ↪  level.
14  - Keep the tool usage sequence and final outcome identical across all
    ↪  variations.
15  - Ensure each variation feels like a realistic scenario in its new
    ↪  context and remains solvable with the same tool operations.
16  - Ensure the question does not contain any tool names or explicit
    ↪  references to the target tools.
17
18  ## Input Format
19  **Original Question**: {ORIGINAL_QUESTION}
```

```
**Target Tools**: {TARGET_TOOLS}
**Tool Descriptions**: {TOOL_DESCRIPTIONS}

## Output Requirements
Generate **{VARIATIONS_COUNT} augmented variations** of the original
↪  question. Each variation should:
1. Maintain the same core goal that requires the target tool(s)
2. Use the exact same tool(s) in the same order with the same final
↪  outcome
3. Apply to a completely different context, scenario, or domain
4. Keep the same complexity level and constraints as the original
5. Feel like a natural, real-world scenario from a different setting
6. Be meaningfully different from the original and other variations in
↪  terms of context only
7. Avoid including any explicit mentions, hints, or references to the
↪  target tool names within the question text

## Output
Provide your response in the following XML format:

<response>
  <analysis>
    <!-- Briefly analyze the original question and target tool(s) to
    ↪  understand the core goal, tool usage pattern, complexity level,
    ↪  and expected outcome, then identify how this can be applied
    ↪  across different domains while maintaining operational
    ↪  consistency -->
  </analysis>
  <variations>
    <!-- Generate {VARIATIONS_COUNT} variations, each with <variation_X>,
    ↪  <context>, and <question> tags -->
    <variation_1>
      <context>
        <!-- Brief description of the new domain/scenario introduced -->
      </context>
      <question>
        <!-- The augmented question that maintains the same target
        ↪  tool(s) usage order, complexity, and outcome but in a
        ↪  different context -->
      </question>
    </variation_1>
    <!-- Continue with variation_2, variation_3, etc. as needed based on
    ↪  number of variations -->
  </variations>
</response>
```

The prompt below is designed to enhance task complexity through the introduction of additional constraints.

```
## Task
Generate **augmented variations** of a given question that maintain the
↪  same target tool(s) usage and context but significantly increase the
↪  complexity and constraints required to solve the problem.

## Objective
Take an existing question and its associated target tool(s), then create
↪  multiple sophisticated variations that:
- Use the same target tool(s) to achieve the core goal while navigating
↪  additional complexity layers
- Maintain the same general context and domain as the original question
- Increase multi-dimensional complexity through realistic constraints,
↪  competing requirements, stakeholder considerations, and
↪  interconnected dependencies
- Embed the tool usage within larger, more complex workflows that require
↪  strategic thinking and coordination
```

```
10  - Demonstrate how the same core tool usage applies under vastly different
    ↪  complexity levels
11
12  ## Guidelines
13  - Introduce realistic constraints such as resource limits, compliance
    ↪  requirements, tight timelines, or stakeholder conflicts
14  - Embed the same tool usage inside a broader workflow that requires
    ↪  coordination across teams or systems
15  - Escalate demands (performance, scalability, risk management) without
    ↪  changing the original domain or context
16  - Ensure each variation targets a different primary complexity angle
    ↪  (organizational, technical, strategic) while preserving tool
    ↪  relevance
17  - Ensure the question does not contain any tool names or explicit
    ↪  references to the target tools.
18
19  ## Input Format
20  **Original Question**: {ORIGINAL_QUESTION}
21  **Target Tools**: {TARGET_TOOLS}
22  **Tool Descriptions**: {TOOL_DESCRIPTIONS}
23
24  ## Output Requirements
25  Generate **{VARIATIONS_COUNT} strategically augmented variations** of the
    ↪  original question. Each variation should:
26  1. Maintain the same core goal that requires the target tool(s) while
    ↪  adding multiple complexity layers
27  2. Keep the same general context and domain as the original question
28  3. Introduce different but interconnected constraints and competing
    ↪  requirements
29  4. Feel like natural, high-stakes, real-world scenarios that
    ↪  professionals encounter
30  5. Be meaningfully different from the original and other variations in
    ↪  terms of complexity
31  6. Include specific details that make the constraints and requirements
    ↪  concrete and actionable
32  7. **Transform step-wise questions**: If the original question contains
    ↪  explicit steps, convert it to a goal-oriented format while
    ↪  maintaining the same tool usage requirements
33  8. Avoid including any explicit mentions, hints, or references to the
    ↪  target tool names within the question text
34
35  ## Output
36  Provide your response in the following XML format:
37
38  <response>
39    <analysis>
40      <!-- Analyze the original question and target tool(s) to understand
         ↪  the core goal, current complexity level, and identify multiple
         ↪  complexity dimensions that can be naturally introduced while
         ↪  maintaining tool relevance and solution feasibility -->
41    </analysis>
42    <variations>
43      <!-- Generate {VARIATIONS_COUNT} variations, each with <variation_X>,
         ↪  <constraints>, and <question> tags -->
44      <variation_1>
45        <constraints>
46          <!-- Specific organizational, stakeholder, or coordination
             ↪  constraints that add realistic complexity -->
47        </constraints>
48        <question>
49          <!-- The complex, organizationally-focused question that
             ↪  maintains the same target tool(s) usage within a more
             ↪  sophisticated workflow -->
50        </question>
51      </variation_1>
```

```
52       <!-- Continue with variation_2, variation_3, etc. as needed based on
         ↪  number of variations -->
53    </variations>
54 </response>
55
```

## D.4 TASK QUALITY ANNOTATION PROMPT

```
1  ## Task
2  Conduct a **Question Quality Assessment** of a tool use question across
   ↪  six key dimensions to ensure it meets high standards for realistic
   ↪  tool usage scenarios.
3
4  ## Objective
5  Analyze the provided tool use question and assess its quality across six
   ↪  primary dimensions:
6  1. **Tool Selection Difficulty** - How challenging it is to determine
   ↪  which tools to use giving all available tools
7  2. **Tool Selection Uniqueness** - How unique and necessary the selected
   ↪  tools are for this specific task giving all available tools
8  3. **Question Quality** - Overall clarity, specificity, and effectiveness
9  4. **Scenario Realism** - How authentic and believable the scenario is
10 5. **Verifiable** - How easy it is to verify the correctness of the final
   ↪  model answer
11 6. **Stability** - How stable the answer will be when requested under
   ↪  different time and geolocation
12
13 ## Assessment Criteria
14
15 ### 1. Tool Selection Difficulty
16 **What to Evaluate**: How difficult it would be for a user to determine
   ↪  which specific tools are needed to solve this question.
17
18 **Rating Guidelines**:
19 - **very easy**: Question explicitly mentions tool names or makes tool
   ↪  selection obvious
20 - **easy**: Tool selection is straightforward with clear indicators
21 - **medium**: Requires some reasoning but tool needs are fairly apparent
22 - **hard**: Requires careful analysis to determine appropriate tools
23 - **very hard**: Requires extensive expertise and deep reasoning to
   ↪  identify the correct tools
24
25 ### 2. Tool Selection Uniqueness
26 **What to Evaluate**: How unique and necessary the selected tools are for
   ↪  accomplishing this specific task, and whether the task can only be
   ↪  completed with these tools in the specified sequence.
27
28 **Rating Guidelines**:
29 - **not unique**: Many alternative tool combinations could accomplish the
   ↪  same task equally well
30 - **somewhat unique**: Some alternative approaches exist, but selected
   ↪  tools offer advantages
31 - **moderately unique**: Selected tools are well-suited, with limited
   ↪  alternative approaches
32 - **quite unique**: Selected tools are particularly well-matched to the
   ↪  task requirements
33 - **highly unique**: Task can only be accomplished effectively with these
   ↪  specific tools in this sequence
34
35 ### 3. Question Quality
36 **What to Evaluate**: Overall quality, clarity, and effectiveness of the
   ↪  question as a realistic user query.
37
38 **Rating Guidelines**:
39 - **very poor**: Unclear, ambiguous, or poorly constructed question
```

```
40  - **poor**: Some clarity issues, missing important context
41  - **average**: Clear and understandable, but could be more specific or
    ↪  engaging
42  - **good**: Well-constructed, clear, specific, and realistic
43  - **excellent**: Exceptionally clear, detailed, engaging, and
    ↪  professionally written
44
45  ### 4. Scenario Realism
46  **What to Evaluate**: How authentic, believable, and true-to-life the
    ↪  described scenario is.
47
48  **Rating Guidelines**:
49  - **unrealistic**: Artificial, contrived, or implausible scenario
50  - **somewhat unrealistic**: Some realistic elements but feels forced or
    ↪  unlikely
51  - **moderately realistic**: Believable scenario with minor authenticity
    ↪  issues
52  - **realistic**: Authentic scenario that represents genuine use cases
53  - **highly realistic**: Completely natural, authentic scenario
    ↪  indistinguishable from real user needs
54
55  ### 5. Verifiable
56  **What to Evaluate**: How easy it is to verify the correctness of the
    ↪  final model answer.
57
58  **Rating Guidelines**:
59  - **hard to verify**: Fully free-form answer that requires extensive
    ↪  human judgment
60  - **somewhat hard**: Mostly subjective answer with some verifiable
    ↪  elements
61  - **moderately verifiable**: Short sentence that can be verified by LLM
    ↪  comparison
62  - **mostly verifiable**: Answer with clear, objective components and some
    ↪  subjective elements
63  - **easy to verify**: Answer can be verified by simple rules, exact
    ↪  matches, or clear success criteria
64
65  ### 6. Stability (1-5 Scale)
66  **What to Evaluate**: How stable and consistent the answer will be when
    ↪  the question is asked under different environmental conditions and
    ↪  system contexts. Consider factors like temporal dependency,
    ↪  geographical variations, operating system differences, network
    ↪  environments, and software version variations.
67
68  **Rating Guidelines**:
69  - **highly unstable**: Answer changes significantly across different
    ↪  conditions (real-time data, location-specific, system-dependent)
70  - **somewhat unstable**: Answer may vary moderately based on
    ↪  environmental or system factors
71  - **moderately stable**: Answer mostly consistent with minor variations
    ↪  due to context
72  - **mostly stable**: Answer remains largely consistent across different
    ↪  conditions
73  - **highly stable**: Answer is completely independent of environmental
    ↪  and system factors
74
75  ## Question Analysis
76
77  ### All Available Tools```
78  {ALL_SERVER_AND_TOOL_INFORMATION}
79  ```
80
81  ### Question Content
82  ```
83  {QUESTION_CONTENT}
84  ```
```

```
### Intended Tool for This Question
```
{INTENDED_TOOL}
```

## Output Requirements

Provide analysis with detailed reasoning BEFORE scores for each of the
↪  six metrics.

## Output
Provide your response in the following XML format:

<response>
  <tool_selection_difficulty>
    <reasoning>
      <!-- Detailed explanation including ambiguity level, domain
      ↪  knowledge required, and alternative solutions giving all
      ↪  available tools -->
    </reasoning>
    <rating><!-- Rating: very easy, easy, medium, hard, very hard
    ↪  --></rating>
  </tool_selection_difficulty>

  <tool_selection_uniqueness>
    <reasoning>
      <!-- Detailed explanation of tool necessity, sequential
      ↪  dependencies, and alternative tool viability giving all
      ↪  available tools -->
    </reasoning>
    <rating><!-- Rating: not unique, somewhat unique, moderately unique,
    ↪  quite unique, highly unique --></rating>
  </tool_selection_uniqueness>

  <question_quality>
    <reasoning>
      <!-- Detailed explanation covering linguistic quality, information
      ↪  architecture, and actionability -->
    </reasoning>
    <rating><!-- Rating: very poor, poor, average, good, excellent
    ↪  --></rating>
  </question_quality>

  <scenario_realism>
    <reasoning>
      <!-- Detailed explanation of industry authenticity, workflow
      ↪  accuracy, and stakeholder behavior -->
    </reasoning>
    <rating><!-- Rating: unrealistic, somewhat unrealistic, moderately
    ↪  realistic, realistic, highly realistic --></rating>
  </scenario_realism>

  <verifiable>
    <reasoning>
      <!-- Detailed explanation of answer format, objective criteria, and
      ↪  ground truth availability -->
    </reasoning>
    <rating><!-- Rating: hard to verify, somewhat hard, moderately
    ↪  verifiable, mostly verifiable, easy to verify --></rating>
  </verifiable>

  <stability>
    <reasoning>
      <!-- Detailed explanation of temporal/geographical/system
      ↪  dependencies and environmental factors -->
```

```
137      </reasoning>
138      <rating><!-- Rating: highly unstable, somewhat unstable, moderately
         ↪  stable, mostly stable, highly stable --></rating>
139    </stability>
140 </response>
141
```

## D.5 TRAJECTORY ANNOTATION PROMPT

```
1  ## Task
2  Conduct a **Response Quality Assessment** of a tool-use conversation
   ↪  across two LLM-scored dimensions, with a third dimension computed
   ↪  automatically outside the LLM.
3
4  ## Objective
5  Analyze the provided conversation and assess its response quality across
   ↪  two primary dimensions scored by the LLM, while reserving an
   ↪  additional tool-call accuracy dimension for automated scoring:
6  1. Completeness - Whether the assistant fully accomplished the user's
   ↪  request end-to-end
7  2. Conciseness - Whether the assistant solved the task using the minimum
   ↪  necessary steps and verbosity
8
9  ## Assessment Criteria
10
11 ### 1. Completeness
12 **What to Evaluate**: Did the assistant fully satisfy the user's goal
   ↪  given the conversation context? Consider whether the assistant:
13 - Executed all required steps end-to-end (including
   ↪  saving/exporting/downloading where applicable)
14 - Provided the final deliverable or a working alternative when blocked
   ↪  (e.g., tool failure with a usable fallback)
15 - Included essential confirmations, paths, or instructions to achieve the
   ↪  outcome
16 - Avoided missing key requirements or leaving the user with unresolved
   ↪  gaps
17
18 **Rating Guidelines**:
19 - very incomplete: Major requirements missing; no usable outcome
20 - incomplete: Some key requirements missing; outcome is not directly
   ↪  usable
21 - partially complete: Core steps attempted; outcome usable only with user
   ↪  effort or missing minor requirements
22 - mostly complete: Meets most requirements; small omissions or minor
   ↪  issues remain
23 - fully complete: All requirements met with a usable outcome delivered
24
25 ### 2. Conciseness
26 **What to Evaluate**: Did the assistant achieve the goal with minimal
   ↪  redundancy and steps? Consider whether the assistant:
27 - Avoided repetitive or unnecessary explanations/tool calls
28 - Used the minimal set of steps/tools to complete the task
29 - Kept language concise while preserving clarity
30
31 **Rating Guidelines**:
32 - very redundant: Excessive repetition or unnecessary steps/tool calls
33 - redundant: Noticeable verbosity or extra steps beyond what's needed
34 - average: Reasonably concise with minor extraneous content
35 - concise: Efficient and to the point with minimal overhead
36 - very concise: Maximally efficient while clear and complete
37
38 ## Response Analysis
39
40 ### Question Content
41 ```
```

```
42   {QUESTION_CONTENT}
43   ```
44
45   ### Intended Tool for This Question
46   ```
47   {INTENDED_TOOL}
48   ```
49
50   ### Conversation History
51   ```
52   {CONVERSATION_HISTORY}
53   ```
54
55   ## Output Requirements
56   - Provide detailed reasoning BEFORE ratings for Completeness and
     ↪  Conciseness
57   - Do NOT score Tool Call Accuracy; include placeholders only
58
59   ## Output
60   Provide your response in the following XML format:
61
62   <response>
63     <completeness>
64       <reasoning>
65         <!-- Evaluate if the assistant delivered an end-to-end usable
          ↪  outcome, addressed all requirements, handled tool failures with
          ↪  alternatives, and provided necessary confirmations/paths. -->
66       </reasoning>
67       <rating><!-- Rating: very incomplete, incomplete, partially complete,
          ↪  mostly complete, fully complete --></rating>
68     </completeness>
69
70     <conciseness>
71       <reasoning>
72         <!-- Evaluate if the assistant minimized redundant
          ↪  steps/explanations, avoided unnecessary tool calls, and kept
          ↪  messaging efficient while clear. -->
73       </reasoning>
74       <rating><!-- Rating: very redundant, redundant, average, concise,
          ↪  very concise --></rating>
75     </conciseness>
76   </response>
```

# E  COST AND ACCESSIBILITY

All models used to build TOUCAN (data generation and annotation) are open-source and can be deployed efficiently using vLLM servers, which substantially reduce inference cost. Our data generation pipeline demonstrates that producing high-quality, end-to-end synthetic data is feasible without relying on proprietary models. In this section, we provide additional guidance on reproducing our results and extending TOUCAN to new MCP servers, considering both standard and resource-constrained computational settings. Table 14 lists the models used to build TOUCAN and their corresponding GPU requirements, along with a set of open-source, resource-efficient alternative models that are fully compatible with the TOUCAN tool-trajectory generation pipeline. These alternative options, when combined with manual review and/or lightweight verification tools, could produce data of comparable quality and difficulty to TOUCAN .

Table 14: Approximate (H100) GPU requirements for the models used at each pipeline stage, as well as lightweight open-source alternatives. GPU requirements are provided for full precision inference, quantized versions of these models would further reduce resource requirements.

| Stage | LLM Used | Approx. GPUs (vLLM, BF16) | Alternative (Smaller LLM) | Approx. GPUs (vLLM, BF16) | Notes |
|---|---|---|---|---|---|
| **Task Synthesis** | Mistral-Small-3.2-24B-Instruct-2506 | 1 | N/A | N/A | Already efficient; runs on single GPU. |
| | DevStral-Small-2505 | 1 | N/A | N/A | Already efficient; runs on single GPU. |
| | GPT-OSS-120B | 4 | GPT-OSS-20B | 1 | Suitable trade-off between performance and compute. |
| | Kimi-K2-Instruct | 2 | – | – | MoE with ∼32B active parameters (≈1T total). |
| | Qwen3-32B-Instruct | 1 | Qwen2.5-7B-Instruct | 1 | Lighter variant preserving coherence for synthesis. |
| **Task Filtering** | Kimi-K2-Instruct | 32 | GPT-OSS-20B | 1 | Best performing open model for filtering. |
| **Trajectory Generation** | GPT-OSS-120B | 4 | GPT-OSS-20B | 1 | Strong performance-compute balance in same agent framework. |
| | Kimi-K2-Instruct | 32 | – | – | Efficient, coherent model for synthesis tasks. |
| | Qwen3-32B-Instruct | 1 | Qwen2.5-7B-Instruct | 1 | Lighter model preserving coherence in trajectory generation. |
| **Trajectory Filtering** | GPT-OSS-120B | 4 | GPT-OSS-20B | 1 | Suitable performanceâ-compute compromise for long trajectories. |

