# OpenReview forum: "TOUCAN: Synthesizing 1.5M Tool-Agentic Data from Real-World MCP Environments"
_ICLR.cc/2026/Conference — Submitted to ICLR 2026_

### Official Review · Reviewer_uq6F · 2025-10-21

**Soundness:** 2
**Presentation:** 2
**Contribution:** 3
**Rating:** 2
**Confidence:** 4

**Summary:**

The paper introduces TOUCAN, a tool-agentic dataset containing 1.5 million training samples synthesized from nearly 500 real-world MCP servers. The authors also describe the details of their construction pipeline, based on the target to generate diverse, realistic and challenging tasks. This includes in overall 5 steps: MCP Server Onboarding, Task Synthesis, Task Filtering, Trajectory Generation, and Rule&LLM-Based Post-Filtering. The analytical and experimental results show that TOUCAN preserves high quality and helps on improving performance on diverse benchmarks based on Qwen2.5 series models.

**Strengths:**

1. This paper introduces the largest agentic dataset with 1.5 Million samples for LLMs.
2. The dataset is generated with real MCP servers, providing realistic feedback that is important for LLM tool calling capability acquisition.
3. The analysis shows the diversity of the dataset, as well as the quality to some extent, based on LLM judges.
4. Experimental results show that fine-tuning with the dataset on Qwen2.5 models improves at least 3% scores in several benchmarks.

**Weaknesses:**

1. The technical contribution is limited. The pipeline introduced is heuristic, where many of previous work has applied similar methods. I cannot find any interesting or new things according to the description. For example, rule&LLM-based quality evaluation have been explored in ToolACE and APIGen, and no any distinct design is found. While the authors emphasize generating realistic, diverse and challenging data, the diversity and difficulty things are not found in the generation pipeline.
2. The experiments are not convincing:
   - Only Qwen2.5 models are tested with the proposed dataset.
   - The experiments are conducted on a selected subset of approximately 119K samples, rather than the full 1.5M dataset. This raises concerns about potential overstatement of their contribution, as the released 1.5M data may not all be of high quality.
   - The improvement is also limited, e.g., only achieving about 3% improvement on Qwen2.5-7B-Instruct with the data. This raises the concerns whether we need so many data samples for one single capability (In my personal experience, 10 - 20 samples per tool are enough for learning the usage of the tool. Given that the total tool amount is about 2k in TOUCAN, it is much less data needed). An ablation study on the data amount is needed.
   - The comparison with the exsting public datasets is needed, not just the statistical level, but experimental level.
   - More analysis on the data quality is needed, showing that the accuracy and difficulty of the data is sufficient.
   - Ablation on the stages proposed is needed. We need evidences proving that the proposed five-stage pipeline is optimized.

**Questions:**

1. What is your unique design in your pipeline?
2. Have you tried using more advanced models for data generation? For example, GPT-5 or Claude 4.

---

> ### Author Response · Authors · 2025-11-21
> **Response for Reviewer uq6F**
>
> We thank you for carefully reading our paper and providing constructive feedback. Below we provide a point-by-point response to your questions and concerns.
>
> – The authors.
>
> ---
>
> > **Comment 1:** The technical contribution is limited. The pipeline introduced is heuristic, where many of previous work has applied similar methods. I cannot find any interesting or new things according to the description. For example, rule&LLM-based quality evaluation have been explored in ToolACE and APIGen, and no any distinct design is found. While the authors emphasize generating realistic, diverse and challenging data, the diversity and difficulty things are not found in the generation pipeline.
>
> We thank the reviewer for the feedback. We agree that several components of our pipeline, such as rule-based filtering and LLM-assisted quality control, are conceptually related to prior work like ToolACE and APIGen. However, our contribution lies not in proposing these components in isolation, but in how they are **instantiated, composed, and applied to the tool-agentic setting under MCP ecosystems**, which introduces new challenges beyond prior API-based or environment-specific pipelines.
>
> First, our pipeline addresses a **fundamentally different problem setting**. While ToolACE and APIGen focus on static APIs in relatively fixed environments, Toucan targets open-ended MCP tool ecosystems, where tools are hosted on heterogeneous servers with evolving schemas, diverse semantics, and non-uniform documentation. This setting introduces challenges such as heterogeneous tool schema and cross-server task composition during task synthesis, which are intrinsic to MCP-based tool-agentic systems and not addressed by prior API-centric pipelines.
>
> Second, our notion of “diversity” and “difficulty” is operationalized through concrete mechanisms, not just claimed. Specifically, diversity and challenge arise from:
>
> - Scaling to 2,000+ tools across 27 domains, significantly broader than prior tool datasets,
> - Explicit synthesis of multi-step, multi-tool trajectories requiring cross-server reasoning,
> - A verifier-driven filtering stage that retains trajectories involving tool failure and recovery behavior, rather than only clean executions,
>
> These design choices result in data that is **structurally more complex than previous datasets**, which is reflected in improved generalization to unseen tools and domains, as shown in the strong performance in benchmarks like $\tau$-Bench and MCP-Universe.
>
> Overall, our contribution is not a reimplementation of prior pipelines, but **a scalable, agent-native data synthesis framework that enables training agentic LLMs on realistic, cross-tool, and cross-domain scenarios**, an area not addressed by existing work. We will further clarify these distinctions in the revised manuscript.
>
> ---
>
> > **Comment 2:** Only Qwen2.5 models are tested with the proposed dataset.
>
> We thank the reviewer for pointing this out. While a significant portion of our analysis focuses on the Qwen2.5 family for controlled comparisons, our dataset is not limited to a single model family and generalizes well across architectures and scales.
>
> To address this concern, we additionally evaluate Toucan on `Llama-3.1-8B-Instruct` and `Llama-3.3-70B-Instruct`, two widely used non-Qwen models. As shown below, fine-tuning with Toucan leads to substantial and consistent improvements across all categories. We integrated these results to `Table 2` in the updated paper.
>
> | Model | BFCL Overall | ST (Non-Live) | ST (Live) | MT | Hallu - Relev | Hallu - Irrelev |
> |------|-------------|---------------|-----------|----|----------------|------------------|
> | Llama-3.1-8B-Instruct | 26.23% | 47.96% | 33.63% | 6.38% | 94.44% | 5.26% |
> | **+ Toucan** | **58.46%** | **83.44%** | **70.68%** | **24.88%** | 77.78% | **64.85%** |
> | Llama-3.3-70B-Instruct | 53.03% | 85.23% | 62.86% | 16.38% | 100.00% | 48.50% |
> | **+ Toucan** | **66.20%** | **85.79%** | **73.48%** | **42.25%** | 77.78% | **68.22%** |
>
> [continued]

---

> > ### Author Response · Authors · 2025-11-21
> > **Response for Reviewer uq6F [Part2]**
> >
> > > **Comment 3:** The experiments are conducted on a selected subset of approximately 119K samples, rather than the full 1.5M dataset. This raises concerns about potential overstatement of their contribution, as the released 1.5M data may not all be of high quality.
> >
> > We appreciate the concern. Our choice of the 119K subset is motivated by **cost efficiency and distributional rebalancing**, not selective cherry-picking.
> >
> > Specifically:
> > 1. **Cost-effectiveness**: Training on the full set is prohibitively expensive for comprehensive performance evaluation. The 119K subset enables reproducible experiments under realistic compute budgets.
> > 2. **Rebalancing, not downsampling**: The subset is a **carefully mixture**, not a random sample, combining different subsets of Toucan datasets to reduce distribution skew.
> >
> > In what follows, we report the BFCL performance (also see `Appendix C.5` in the revised version). While Toucan Full slightly outperforms the SFT subset in terms of overall score, it achieves a lower score on Hallucination–Irrelevance. Notably, the relevance mixture in Toucan-SFT is more effective at reducing hallucinations, indicating that careful data mixing plays a critical role in mitigating spurious tool usage.
> >
> > | Model | BFCL Overall | ST (Non-Live) | ST (Live) | MT | Hallu - Relev | Hallu - Irrelev |
> > |------|--------------|---------------|-----------|----|----------------|------------------|
> > | Qwen2.5-14B-Instruct (FC) | 57.69% | 83.38% | 73.70% | 19.75% | 83.33% | 68.46% |
> > | **+ Toucan-SFT (119K)** | 65.09% | **85.42%** | **76.01%** | 35.25% | 72.22% | **75.96%** |
> > | **+ Toucan Full** | **65.17%** | 84.90% | 74.63% | **39.13%** | **83.33%** | 68.71% |
> >
> > ---
> >
> > > **Comment 4:** The improvement is also limited, e.g., only achieving about 3% improvement on Qwen2.5-7B-Instruct with the data. This raises the concerns whether we need so many data samples for one single capability (In my personal experience, 10 - 20 samples per tool are enough for learning the usage of the tool. Given that the total tool amount is about 2k in TOUCAN, it is much less data needed). An ablation study on the data amount is needed.
> >
> > We appreciate the reviewer’s concern. First, we would like to clarify that a 3% absolute improvement on Qwen2.5-7B-Instruct is not marginal, given the strong baseline performance of this model. On other model families, the improvement is significantly larger. For example, on **Llama-3.1-8B-Instruct**, adding Toucan increases the overall score from 26.23% to 58.46%, representing a **~32%** absolute improvement (see response for Comment 1).
> >
> > In addition, we have conducted an **explicit data scale ablation** (sampling from 20K to 119K) to address this question. Our results (also see `Appendix C.6`) show consistent gains with increasing data scale, especially on multi-turn. In addition, we observe diminishing returns and near-saturation behavior after scaling to more than 80K data. This aligns with typical scaling behavior in instruction tuning and supports the idea that our rebalanced subset already offers a cost–performance sweet spot.
> >
> > | Dataset | BFCL Overall | ST (Non-Live) | ST (Live) | MT | Hallu-Relev | Hallu-Irrelev | $\tau$-Airline @1 | $\tau$-Retail @1 |
> > |--------|---------------|---------------|-----------|----|-------------|---------------|-----------|-----------|
> > | Qwen2.5-32B-Instruct (FC) | 61.73% | 85.58% | 76.01% | 26.38% | 72.22% | 72.68% | 26.00% | 51.52% |
> > | Toucan-SFT-20K | 68.21% | 88.52% | 74.99% | 42.50% | 83.33% | 74.73% | 27.75% | 47.50% |
> > | Toucan-SFT-40K | 68.82% | 86.77% | 77.30% | 43.50% | 77.78% | 76.62% | 28.50% | 53.37% |
> > | Toucan-SFT-60K | 68.55% | 86.71% | 77.08% | 43.12% | 83.33% | 75.87% | 28.25% | 56.30% |
> > | Toucan-SFT-80K | 69.62% | 87.02% | 77.65% | 45.25% | 77.78% | 77.23% | 30.00% | 58.26% |
> > | Toucan-SFT-100K | 69.83% | 86.44% | 78.76% | 45.25% | 77.78% | 77.91% | 28.00% | 56.66% |
> > | Toucan-SFT-119K | **70.45%** | 87.12% | **78.90%** | **46.50%** | 77.78% | **78.10%** | 29.00% | 55.65% |
> >
> > [continued]

---

> > > ### Author Response · Authors · 2025-11-21
> > > **Response for Reviewer uq6F [Part3]**
> > >
> > > > **Comment 5:** The comparison with the exsting public datasets is needed, not just the statistical level, but experimental level.
> > >
> > > We appreciate the reviewer’s suggestion and agree that experimental comparisons are important. To address this, we include a controlled comparison in `Appendix C.7` with Nemotron-SFT (tool subset) under a similar data scale (Table 1), using the same base model (Qwen2.5-14B-Instruct).
> > >
> > > While Nemotron-SFT achieves comparable performance on BFCL Overall, **Toucan shows substantially stronger performance** on tool-agentic benchmarks, especially τ-Bench and τ2-Bench, which better reflect multi-tool reasoning and cross-domain generalization.
> > >
> > > | Model | BFCL Overall | ST Non-Live | ST Live | MT | Hallu-Relev | Hallu-Irrelev | τ-Bench Avg | Airline | Retail | τ2-Bench Avg | Airline | Retail | Telecom |
> > > |-------|-------------|-------------|---------|----|-------------|---------------|------------|---------|--------|---------------|---------|--------|---------|
> > > | Qwen2.5-14B + **Toucan** | 65.09% | 85.42% | 76.01% | 35.25% | 72.22% | 75.96% | **35.24%** | 22.00% | 48.48% | **30.43%** | 22.00% | 49.10% | 20.18% |
> > > | Qwen2.5-14B + **Nemotron-SFT (tool subset)** | 65.64% | 85.02% | 81.83% | 30.00% | 66.67% | 85.45% | 24.38% | 18.00% | 30.76% | 20.23% | 16.00% | 36.80% | 7.90% |
> > >
> > > ---
> > >
> > > > **Comment 6:** More analysis on the data quality is needed, showing that the accuracy and difficulty of the data is sufficient.
> > >
> > > We appreciate the reviewer’s suggestion. We have already provided a detailed data quality analysis in Section 3.3, where we evaluate Toucan instances using LLM-as-a-judge and show consistently high quality across multiple criteria. In the revised version, we additionally include more case studies to further illustrate the accuracy and error recovery behaviors in the trajectories.
> > >
> > > ---
> > >
> > > > **Comment 7:** Ablation on the stages proposed is needed. We need evidences proving that the proposed five-stage pipeline is optimized.
> > >
> > > We thank the reviewer for this suggestion. Stages 1–4 of our pipeline (server onboarding, task synthesis, trajectory generation, and verifier-based correction) are structural components required to construct a complete tool-agentic dataset. Removing any of them would result in missing or unusable trajectories.
> > >
> > > However, we agree that Stage 5 (Rule & LLM-based filtering) is the most suitable stage for controlled ablation. We therefore conduct an ablation study comparing datasets with and without this filtering step, while keeping all other stages and the total data size identical. We observed that the overall peformance on BFCL V3, as well as the multi-turn and irrelevance setting benefit from filtering, which confirms the value of **including an automated process to filter-out low-quality trajectories** in data generation pipelines.}
> > >
> > > | Model | BFCL Overall | ST (Non-Live) | ST (Live) | MT | Hallu - Relev | Hallu - Irrelev |
> > > |------|---------------|---------------|-----------|----|---------------|-----------------|
> > > | Qwen2.5-14B-Instruct (FC) | 57.69% | 83.38% | 73.70% | 19.75% | 83.33% | 68.46% |
> > > | **Toucan – No Filtering** | 62.60% | **86.83%** | 72.01% | 32.25% | 77.78% | 67.01% |
> > > | **Toucan – Filtered (Full 5 stages)** | **65.09%** | 85.42% | **76.01%** | **35.25%** | 72.22% | **75.96%** |
> > >
> > > We further note that we have already conducted ablation analyses on pipeline extensions in Section 4.3, which demonstrate that each extension contributes positively to the final performance.
> > >
> > > ---
> > >
> > > > **Question 1:** What is your unique design in your pipeline?
> > >
> > > As mentioned in the previous response, our unique contribution lies in a MCP-native, end-to-end data construction pipeline for agent training, rather than proposing isolated heuristic components. These designs enable large-scale, cross-domain, multi-tool trajectory generation from any MCP servers, an aspect not addressed in prior API-centric pipelines.
> > >
> > > ---
> > >
> > > > **Question 2:** Have you tried using more advanced models for data generation? For example, GPT-5 or Claude 4.
> > >
> > > We have not used GPT-5 or Claude-4 for large-scale generation due to budget constraints and their **license restrictions** associated with distilled and derived model output usage. However, **our pipeline is model-agnostic**: it is fully compatible with both open-source and closed-source models, and our released codebase in supplementary materials supports easy integration of alternative backbone models. This allows future users to plug in more advanced models as they become available, without changing the pipeline design.

---

> > > > ### Author Response · Authors · 2025-11-25
> > > > **Follow-Up on Author Response**
> > > >
> > > > Dear reviewer,
> > > >
> > > > I am writing to follow up on our author response submitted on November 21st. With the December 2nd revision deadline approaching, we are concerned that we may not have enough time to address any additional questions you might have.
> > > >
> > > > We would greatly appreciate your engagement in the discussion at your earliest convenience.
> > > >
> > > > Thank you for your time and efforts. We look forward to hearing from you.

---

> > > > > ### Comment · Reviewer_uq6F · 2025-11-26
> > > > >
> > > > > Thank you for providing the additional experimental results! Incorporating these results into the paper significantly strengthens its validity, and I hope the authors will revise the manuscript accordingly. Regarding the technical contribution, I still maintain my original assessment. Transitioning from a “static API” setup to an “open-ended MCP tool ecosystem” reflects more of an engineering effort than a substantive research contribution, and thus should not be counted as a major technical novelty. Moreover, although the authors emphasize the “open-ended MCP tool ecosystem” setting, I do not see any novel or distinctive design in the data construction pipeline considering this setting aside from the MCP server collection. I have raised my score to 4 based on the added experiments.

---

> > > > > > ### Author Response · Authors · 2025-11-26
> > > > > >
> > > > > > Dear Reviewer,
> > > > > >
> > > > > > Thank you for your thoughtful feedback and for taking the time to reassess our work. We appreciate your acknowledgement of the additional experimental results and are glad to hear that they have strengthened the soundness of the manuscript. As noted, **our updated submission already incorporates these results, along with several revisions intended to address earlier concerns**.
> > > > > >
> > > > > > Regarding the novelty of the construction pipeline, we would like to respectfully clarify how it aligns with the conference’s evaluation criteria. The objective of our work is to provide a principled, extensible, and reproducible methodology for constructing large-scale tool-use trajectories that support the development and study of LLM tool-use in agentic settings. Within this objective, **the transition from a static API setup to an open-ended MCP-based pipeline is not merely an engineering refactoring, but a motivated design decision that enables reliable data-generation of trajectories**. MCP offers extensibility, modularity, and interoperability that allow the pipeline to scale to diverse tool ecosystems, these properties are critical to build better LLM agents.
> > > > > >
> > > > > > Beyond this architectural foundation, the pipeline is designed to explicitly optimize for diversity, completeness, and difficulty in the resulting trajectories. This includes support for heterogeneous tool-use scenarios that would be difficult to construct, standardize, or scale through existing approaches. These design components, as well as the accompanying empirical analysis, directly support the claims made in the paper regarding the need for richer and more representative datasets for tool-use agents.
> > > > > >
> > > > > > **Importantly, the construction pipeline is only one part of the contribution. The Toucan dataset itself has clear significance and value for the ICLR community and beyond**. It provides a large-scale, open, and reproducible training resource at a time when high-quality data for tool-augmented models is increasingly essential but remains largely inaccessible due to closed-source infrastructure. Toucan helps bridge this gap by offering a transparent, extensible dataset that can be used by researchers, practitioners, and open-source contributors alike. We kindly ask that this broader audience and impact be taken into account when assessing the work.
> > > > > >
> > > > > > We appreciate your engagement and your updated score, and we hope that the revised manuscript more clearly conveys the novelty, motivation, and contribution of our work in line with the conference guidelines.

---

> > > > > > > ### Comment · Reviewer_uq6F · 2025-11-27
> > > > > > >
> > > > > > > Thank you for your response. I appreciate seeing the updated version.

---

> > > > > > > > ### Author Response · Authors · 2025-11-27
> > > > > > > >
> > > > > > > > Dear Reviewer,
> > > > > > > >
> > > > > > > > Thank you for your response and for reviewing the updated version of our manuscript. Given the substantial revisions we made in direct response to your earlier concerns, we would appreciate more detailed feedback and assessment following the rebuttal. In particular, we would like to better understand the basis of your initial evaluation:
> > > > > > > >
> > > > > > > > **Soundness: 2 (fair)**
> > > > > > > >
> > > > > > > > Your review highlighted weaknesses in soundness, which we found extremely insightful and helpful. As you saw in our rebuttal response, we worked diligently to address them by extending our evaluation and conducting several additional ablations, all included in the revised manuscript. We understand the increment to overall score is directly related to this point, please let us know otherwise.
> > > > > > > >
> > > > > > > > **Presentation: 2 (fair)**
> > > > > > > >
> > > > > > > > You also assigned a low score for presentation, but the specific issues underlying this score were not detailed. Any clarification you can provide would help us further improve the clarity and readability of the manuscript.
> > > > > > > >
> > > > > > > > **Contribution: 3 (good)**
> > > > > > > >
> > > > > > > > Given that you assessed the contribution as good, we found your after-rebuttal comment regarding "limited technical contribution" somewhat unclear. We would appreciate clarification.
> > > > > > > >
> > > > > > > > We are in the best disposition to clarify an discuss any additional doubts you may have.
> > > > > > > >
> > > > > > > > Thanks,

---

> > > > > > > > > ### Comment · Reviewer_uq6F · 2025-11-28
> > > > > > > > >
> > > > > > > > > I have raised the final score according to the updated experiments, and I apologize for previously forgetting to revise the soundness and presentation sections. However, I still do not find the technical contribution convincing. The contribution score is primarily attributed to the open-source release (which I really appreciate—releasing the datasets is highly valuable to the community). That said, I do not see any novel aspects in the method description regarding the claimed “transition from a static API setup to an open-ended MCP-based pipeline” or the ability to “optimize for diversity, completeness, and difficulty.” The five stages and the extension steps appear to be standard operations commonly applied in prior work, without clear methodological innovation.

---

### Official Review · Reviewer_CckG · 2025-10-29

**Soundness:** 3
**Presentation:** 3
**Contribution:** 3
**Rating:** 6
**Confidence:** 4

**Summary:**

This work introduces TOUCAN, the largest publicly available tool-agentic dataset to date, comprising 1.5 million trajectories synthesized from nearly 500 real-world MCP servers. The TOUCAN Generation Pipeline consists of five stages: 1. MCP Server Onboarding 2.Task Synthesis 3.Task Filtering 4. Trajectory Generation 5. Rule&LLM-Based Post-Filtering, and 3 distinct procedures post-core pipeline (Steps 1 to 5) to generate new instances targeting specific objectives:1. Irrelevance, 2. Persona-based Diversification, 3. Multi-Turn.

Empirically, models fine-tuned on TOUCAN significantly outperform both open- and closed-source baselines across diverse benchmarks.

**Strengths:**

- The paper is well-organized and easy to follow, with clear presentation of dataset statistics, filtering criteria, and evaluation metrics.

- The data generation pipeline is described in detail, systematically addressing the common challenges in tool-calling dataset construction.

- The work is highly valuable, I believe tool-use capability is a crucial step for LLMs toward AIGC.

**Weaknesses:**

- Although the reported performance gains are significant, there remains a gap to the latest SOTA. For instance, xLAM-2-70B-fc-r achieves 75.38 on the BFCL-V3 multi-turn benchmark, higher than the TOUCAN-tuned counterparts.

- The 495 MCP servers used may still be insufficient to cover the full spectrum of real-world scenarios. Expanding the dataset to include more domains.

**Questions:**

I briefly take a look at the released dataset and noticed that many MCP servers contain overlapping or functionally similar tools (e.g., search tools).
Did the authors consider deduplication of MCP servers during the MCP Server Onboarding stage to reduce redundancy?

---

> ### Author Response · Authors · 2025-11-21
> **Response for Reviewer CckG**
>
> We thank you for carefully reading our paper and providing constructive feedback. Below we provide a point-by-point response to your questions and concerns.
>
> – The authors.
>
> > **Comment 1:** Although the reported performance gains are significant, there remains a gap to the latest SOTA. For instance, xLAM-2-70B-fc-r achieves 75.38 on the BFCL-V3 multi-turn benchmark, higher than the TOUCAN-tuned counterparts.
>
> We appreciate the reviewer’s observation. We would like to clarify key differences in training data composition that directly impact performance on BFCL-V3. We use `xLAM-2-32B-fc-r` as the reference point, rather than the 70B variant, since this is the largest Qwen model we used in our experiments.
>
> A critical difference lies in the **training data composition**. APIGent-MT-5K, the dataset used to train `xLAM-2-32B-fc-r`, is composed entirely of multi-turn trajectories. This design naturally boosts performance on the **multi-turn split** of BFCL-V3. Because BFCL-V3 reports a combined score across both single-turn and multi-turn tasks, a dataset optimized exclusively for multi-turn interactions also maximizes overall performance. This is reflected in the per-category results below: `xLAM-2-32B-fc-r` achieves an exceptionally high multi-turn score. That said, `xLAM-2-32B-fc-r` is not exceptional in all categories, our fine-tuned Qwen2.5-32B baseline outperforms it on `single-turn Live (AST)` as well as the `Irrelevance` group.
>
> | Model               | Non-live (AST) | Live (AST) | Multi Turn | Relevance | Irrelevance|
> | --------            | --------       | --------   | --------   | --------  | ------     |
> | `xLAM-2-32b-fc-r`   | **89.50**      | 73.79      | **66.38**  | **83.33** | 76.25      |
> | `Qwen2.5-32B+Toucan`| 87.12          | **78.990** | 46.50      | 77.78     |**78.10**   |
>
> More importantly, **APIGent-MT-5K is generated using the same tool environments as τ-Bench**:
>
> "Data Collection Procedure. We source APIs implemented as Python functions from τ-bench. Among these, we have 15 ‘read’ and 13 ‘write’ APIs across both domains. τ-bench is accompanied with detailed policies and domain rules in two settings - Retail and Airline which we use as guideline policies."* [(Prabhakar et al. , 2025)](https://arxiv.org/pdf/2504.03601)
>
> Also noted in `Table 1` of our paper. This means the resulting **evaluation is largely in-distribution** with respect to its training data.
>
> In contrast, Toucan is constructed using out-of-distribution tool ecosystems relative to these benchmarks. Our models are therefore evaluated on unseen tools and unseen domains, making the generalization setting significantly more challenging and realistic.
>
> Therefore, while the results of `xLAM-2-32B-fc-r` are impressive, the two setups differ fundamentally in data distribution and task assumptions, and should not be interpreted as directly comparable. We believe our results more accurately reflect model generalization to real-world, unseen tool environments.
>
> ---
>
> > **Comment 2:** The 495 MCP servers used may still be insufficient to cover the full spectrum of real-world scenarios. Expanding the dataset to include more domains.
>
> We appreciate the reviewer’s point. While no finite set of MCP servers can fully capture the diversity of real-world tools, our goal is to **ensure broad generalization** rather than exhaustive coverage. In this regard, the 495 MCP servers we use already span a wide range of capabilities. Moreover, some of the domains appearing in MCP Universe are OOD (see `Appendix B.4`) with respect to our training servers. Despite this distribution shift, our models continue to perform strongly, demonstrating that the dataset is sufficient to support robust generalization beyond its immediate coverage.
>
> ---
>
> > **Question 1:** I briefly take a look at the released dataset and noticed that many MCP servers contain overlapping or functionally similar tools (e.g., search tools). Did the authors consider deduplication of MCP servers during the MCP Server Onboarding stage to reduce redundancy?
>
> We appreciate the reviewer’s observation. While it is true that some MCP servers provide functionally similar capabilities (e.g., different search tools), our pipeline addresses redundancy at the task level rather than at the server level. During the first stage of task synthesis (`Section 3.1`, Stage 1), we perform embedding-based near-deduplication to ensure that the generated tasks remain diverse regardless of which specific tools or servers they originate from. This approach allows us to retain a broad variety of tool implementations while avoiding redundancy in the resulting training data.

---

> > ### Author Response · Authors · 2025-11-25
> > **Follow-Up on Author Response**
> >
> > Dear reviewer,
> >
> > I am writing to follow up on our author response submitted on November 21st. With the December 2nd revision deadline approaching, we are concerned that we may not have enough time to address any additional questions you might have.
> >
> > We would greatly appreciate your engagement in the discussion at your earliest convenience.
> >
> > Thank you for your time and efforts. We look forward to hearing from you.

---

> > > ### Comment · Reviewer_CckG · 2025-11-27
> > >
> > > I appreciate the authors’ detailed response.  I am keeping my initial rating.

---

> > > > ### Author Response · Authors · 2025-11-27
> > > >
> > > > Dear Reviewer,
> > > >
> > > > Thank you for taking the time to read our rebuttal. Since you have opted to keep your initial rating, we would appreciate clarification on which aspects of your original concerns you still find unresolved.
> > > >
> > > > Second, we would like to inform you that for our updated paper submission, we have extended our evaluation and conducted four additional ablation studies, all of which are included in the revised manuscript. These additional experiments significantly strengthen the soundness of our evaluation and results. It would be greatly appreciated if you could review these new results and reconsider your score.
> > > >
> > > > Thank you,

---

### Official Review · Reviewer_tfUW · 2025-11-01

**Soundness:** 3
**Presentation:** 3
**Contribution:** 3
**Rating:** 6
**Confidence:** 4

**Summary:**

This paper presents TOUCAN, a dataset of 1.5M tool-agentic trajectories from 495 real-world MCP servers. Unlike prior work using simulated responses, TOUCAN uses authentic tool execution. The systematic pipeline includes task synthesis, quality filtering, trajectory generation, and three extensions for irrelevance, diversification, and multi-turn dialogues. Models fine-tuned on TOUCAN outperform larger baselines on BFCL V3 and MCP-Universe benchmarks.

**Strengths:**

1. TOUCAN is the largest open-source tool-agentic dataset, using real MCP servers with authentic tool execution rather than simulated responses. The coverage of 495 servers across diverse domains addresses a critical gap in permissively licensed training data.

2. The five-stage pipeline with rigorous filtering (LLM-based quality assessment, rule-based validation) and three thoughtful extensions (irrelevance handling, persona-based diversification, multi-turn conversations) demonstrates careful engineering. The ablation study validates each component's contribution.

3. Extensive documentation including prompts, schemas, hyperparameters, and detailed appendices. The modular pipeline design allows future extensions and customization.

**Weaknesses:**

1. Excluding servers requiring authentication (reducing 2,800 to 495 servers) systematically removes widely-used production services like GitHub, Notion, and Slack. This likely underrepresents enterprise workflows and authenticated API interactions common in real deployments. The impact on domain coverage and practical applicability needs better characterization.

2.  LLM-based quality assessment using Kimi-K2 shows only 0.264 Pearson correlation with human annotations on 50 samples. This small sample size and modest correlation raise concerns about annotation reliability, especially since similar models are used for both annotation and training.

3. All benchmarks may share tool characteristics with the training set. The paper lacks evaluation on completely unseen tool ecosystems or domains, making it unclear whether models learn general tool-use capabilities or memorize specific patterns.

**Questions:**

1.  What is the estimated domain coverage loss from excluding authenticated servers? Have you quantified how many critical real-world use cases are missing?

2.  What is the inter-annotator agreement among human evaluators? With only 0.264 correlation, how confident are you in the quality scores? Have you considered ensemble approaches with multiple judges?

3. Have you evaluated on completely unseen tool ecosystems (domains not in training)? What about zero-shot performance on novel tool combinations?

4. How do you ensure test sets from benchmarks were not included through overlapping MCP servers or similar task formulations?

5. Since you filter out tool failures, how do models learn robust error recovery? Could you include failure cases with recovery strategies?

---

> ### Author Response · Authors · 2025-11-21
> **Response for Reviewer tfUW**
>
> We thank you for carefully reading our paper and providing constructive feedback. Below we provide a point-by-point response to your questions and concerns.
>
> – The authors.
>
> ---
>
> > **Comment 1:** Excluding servers requiring authentication (reducing 2,800 to 495 servers) systematically removes widely-used production services like GitHub, Notion, and Slack. This likely underrepresents enterprise workflows and authenticated API interactions common in real deployments. The impact on domain coverage and practical applicability needs better characterization.
>
> We thank the reviewer for this important point. While we filter out servers that require authentication during data construction to ensure reproducibility, accessibility, and to reduce the burden on MCP providers, we emphasize that this does not significantly limit the enterprise relevance or real-world applicability of our models.
>
> Specifically, our evaluation on the MCP Universe benchmark demonstrates that models fine-tuned on Toucan generalize well to workflows involving enterprise-grade tools that were not observed during training. MCP Universe includes commonly used production MCP servers such as Google Maps, Google Search, GitHub, and Blender, all of which require interacting with realistic (paid) APIs and complex tool interfaces. Despite never seeing these authenticated servers during training, our models exhibit strong performance on these tasks, indicating robust cross-tool and cross-domain generalization.
>
> Therefore, although Toucan omits authenticated servers from its training corpus, our evaluation provides direct evidence that the resulting models still transfer effectively to realistic enterprise workflows and authenticated API interactions.
>
> ---
>
> > **Comment 2:** LLM-based quality assessment using Kimi-K2 shows only 0.264 Pearson correlation with human annotations on 50 samples. This small sample size and modest correlation raise concerns about annotation reliability, especially since similar models are used for both annotation and training.
>
> In response to the reviewer’s concern, we highlight three clarifications and updates:
>
> First, the Pearson correlation between the two human annotators is **0.5028**, indicating that even human judgments exhibit moderate agreement. This suggests that the lower correlations observed with LLMs are partly due to the **inherent subjectivity of the annotation task** (e.g., tool selection uniqueness, scenario realism), rather than model unreliability. This interpretation is consistent with prior findings that LLMs often achieve moderate agreement with human annotators on similarly subjective evaluation tasks [(Pavlović et al., 2024)](https://aclanthology.org/2024.nlperspectives-1.11.pdf?utm_source=chatgpt.com). We have updated `Appendix C.1` to report human annotators score.
>
> Second, while a Pearson correlation of 0.264 may appear modest in isolation, **Kimi-K2 ranks second overall among all evaluated models** (see `Apppendix C.1`) and first among open-source models in our comparison, demonstrating competitive performance under a challenging and subjective evaluation setup.
>
> Third, we clarify that LLM-based annotation scores are not used as training signals. Instead, they are used **solely as a pre-filtering mechanism** to remove clearly low-quality task samples before data mixture. Therefore, concerns about annotation–training contamination or circular evaluation may not apply.
>
> In addition, as suggested by the reviewer, we further explored ensemble approaches. The results are summarized below. Notably, the fully open-source ensemble of Kimi-K2 + Qwen3-235B-A22B slightly outperforms Kimi-K2 alone, indicating that **model ensembling could partially mitigate annotation noise**.
>
> | Rank | Ensemble Combination | Avg Pearson |
> |------|---------------------|-------------|
> | 1 | GPT-4.1 + Grok-3-Mini | **0.3748** |
> | 2 | DeepSeek-V3 + GPT-4.1 | 0.3478 |
> | 3 | GPT-4.1 + Qwen3-32B | 0.3477 |
> | 4 | GPT-4.1 + Kimi-K2 | 0.3411 |
> | 5 | GPT-4.1 + Qwen3-235B-A22B | 0.3381 |
> | 6 | DevStral-Small + GPT-4.1 | 0.3143 |
> | 7 | Grok-3-Mini + Kimi-K2 | 0.2989 |
> | 8 | Kimi-K2 + Qwen3-235B-A22B | 0.2733 |
> | 9 | DeepSeek-V3 + Kimi-K2 | 0.2723 |
> | 10 | DevStral-Small + Kimi-K2 | 0.2714 |
>
> [continued]

---

> > ### Author Response · Authors · 2025-11-21
> > **Response for Reviewer tfUW [Part2]**
> >
> > > **Comment 3:** All benchmarks may share tool characteristics with the training set. The paper lacks evaluation on completely unseen tool ecosystems or domains, making it unclear whether models learn general tool-use capabilities or memorize specific patterns.
> >
> > We appreciate the reviewer’s concern regarding potential overlap between training and evaluation tool ecosystems. We would like to clarify that our evaluations are explicitly designed to test **generalization to unseen tools and domains**. For example, in Table 3, the tools used in the domain setups of **both τ-Bench and τ2-Bench are entirely excluded** from our training data, yet we still observe a consistent and significant performance improvement after Toucan fine-tuning, indicating effective cross-tool generalization rather than memorization.
> >
> > In addition, as detailed in Appendix B.4, MCP Universe includes four domains that are **entirely out-of-distribution (OOD)** with respect to Toucan: location navigation, repository management, 3D design, and web search. These domains are absent from our training corpus because their underlying MCP servers require paid APIs and authentication, and were therefore intentionally excluded during data construction.
> >
> > As a result, our models are evaluated on tools they have never encountered during training, both at the tool level and at the domain level. The strong performance of Toucan-finetuned models on these MCP Universe tasks provides direct evidence that our models **learn generalizable tool-use behaviors**, rather than memorizing patterns tied to specific tools or domains.
> >
> > We have added a detailed domain overlap analysis in Appendix B.4 of the revised manuscript to further clarify this point.
> >
> > ---
> >
> > > **Question 1:** What is the estimated domain coverage loss from excluding authenticated servers? Have you quantified how many critical real-world use cases are missing?
> >
> > We appreciate the reviewer’s concern. However, quantifying which “critical real-world use cases” are missing is not currently feasible, as there is no standardized or comprehensive taxonomy of real-world tool-based workflows or enterprise use cases to benchmark against. Instead, we rely on empirical evaluation across diverse, realistic benchmarks, and T-SNE plots in Figure 12 in the appendix. Please see our response to Comment 1 for further evidence and discussion. We appreciate any further suggestions by the reviewer on the taxonomy!
> >
> > ---
> >
> > > **Question 2:** What is the inter-annotator agreement among human evaluators? With only 0.264 correlation, how confident are you in the quality scores? Have you considered ensemble approaches with multiple judges?
> >
> > We have addressed this question in our response to Comment 2.
> >
> > ---
> >
> > > **Question 3:** Have you evaluated on completely unseen tool ecosystems (domains not in training)? What about zero-shot performance on novel tool combinations?
> >
> > Yes, all our evaluations were conducted on tool ecosystems that are not seen during training. In particular, as discussed in our response to [tfUW.3], MCP Universe contains four domains that are entirely out-of-distribution (OOD) with respect to Toucan since the underlying MCP servers require paid APIs and authentication and are therefore excluded from our training corpus. This ensures that our evaluation tests generalization at both the tool and domain levels.
> >
> > Regarding the second question, τ and τ2 explicitly requires multi-tool usage, and MCP Universe also includes tasks in which models must coordinate interactions with multiple external tools within a single trajectory. Both evaluations show that Toucan consistently improves models’ zero-shot performance on multi-tool tasks. This is expected, as our pipeline is designed to synthesise trajectories that combine tools across different MCP servers and featured servers (see Section 3.1, Stage 2), thereby encouraging generalization to novel tool compositions rather than memorization.
> >
> > ---
> >
> > > **Question 4:** How do you ensure test sets from benchmarks were not included through overlapping MCP servers or similar task formulations?
> >
> > To minimize potential overlap, we source MCP servers for training from repositories and providers that differ from those used by the benchmark suites. In practice, a small number of highly generic infrastructure servers (e.g., time, fetch) are widely adopted across the ecosystem and therefore difficult to fully avoid. However, these servers provide only low-level, domain-agnostic functionalities and do not encode task- or benchmark-specific logic. Beyond such minimal cases, we carefully check for server- and tool-schema similarities with MCP Universe and avoid overlapping servers whenever possible, thereby substantially reducing the risk of leakage.
> >
> > [continued]

---

> > > ### Author Response · Authors · 2025-11-21
> > > **Response for Reviewer tfUW [Part3]**
> > >
> > > > **Question 5:** ince you filter out tool failures, how do models learn robust error recovery? Could you include failure cases with recovery strategies?
> > >
> > > We appreciate the question. To clarify, our filtering only removes trajectories in which **all** tool calls fail, we have updated `Section 3.1, Stage 5` of our paper to make it clear. We do retain trajectories where the model makes one or more mistakes but ultimately succeeds after issuing corrective tool calls. These partial-failure trajectories naturally contain examples of recovery strategies, such as reformatting arguments, retrying with correction issues, or switching to an more appropriate tool.

---

> ### Author Response · Authors · 2025-11-25
> **Follow-Up on Author Response**
>
> Dear reviewer,
>
> I am writing to follow up on our author response submitted on November 21st. With the December 2nd revision deadline approaching, we are concerned that we may not have enough time to address any additional questions you might have.
>
> We would greatly appreciate your engagement in the discussion at your earliest convenience.
>
> Thank you for your time and efforts. We look forward to hearing from you.

---

### Official Review · Reviewer_Rknv · 2025-11-01

**Soundness:** 2
**Presentation:** 3
**Contribution:** 2
**Rating:** 4
**Confidence:** 4

**Summary:**

The paper introduces TOUCAN, a large-scale tool-agentic dataset comprising over 1.5 million trajectories from 495 real-world Model Context Protocols (MCPs). The goal of TOUCAN is to address the limitations of current open-source datasets by providing more diverse and realistic tool-agentic interactions. It presents a comprehensive data generation pipeline involving task synthesis, trajectory generation, and rigorous filtering for high-quality outputs. The paper also highlights improvements in model performance when fine-tuned on TOUCAN, particularly on benchmarks such as BFCL V3 and MCP-Universe.

**Strengths:**

1.  TOUCAN offers a large-scale, high-quality dataset that provides diverse real-world tool-agentic interactions, filling a significant gap in available open-source datasets for training agentic LLMs.

2. The paper demonstrates clear improvements in model performance after fine-tuning on TOUCAN, outperforming existing models on multiple benchmarks. This provides evidence for the practical impact of the dataset.

3. TOUCAN covers a wide variety of domains, tools, and interaction patterns, making it a versatile resource for various LLM-based tasks.

**Weaknesses:**

1. The method relies heavily on large language models and teacher models to generate tool-agentic interactions, which is computationally expensive. This may limit its accessibility for research groups without sufficient resources.

2. Models trained on this dataset may be overfitted to the 2000 tools. How to extend the models to other domains with different tools? These paper may have limited usefulness to practical tasks.

3. The pipeline described for task generation and filtering is similar to previously proposed methods, such as in [a] (Multi-modal Agent Tuning: Building a VLM-Driven Agent for Efficient Tool Usage, ICLR 2025). While the data volume is significantly larger in this manuscript, the methodological novelty may be questioned. The relationship and differences should be discussed in the manuscript.

4. The dataset involves more than 2,000 tools. It is unclear how the inclusion of so many tools adds unique value to the dataset. A discussion of the redundancy and actual diversity of these tools would help clarify whether they introduce meaningful variety or if many of them are effectively duplicates.

**Questions:**

see the weaknesses.

---

> ### Author Response · Authors · 2025-11-21
> **Response for Reviewer Rknv [Part1]**
>
> We thank you for carefully reading our paper and providing constructive feedback. Below we provide a point-by-point response to your questions and concerns.
>
> – The authors.
>
> ---
>
> > **Comment 1:** The method relies heavily on large language models and teacher models to generate tool-agentic interactions, which is computationally expensive. This may limit its accessibility for research groups without sufficient resources.
>
> We appreciate the reviewer's concern about the accessibility of our work for researchers with limited computational resources. While our data construction pipeline indeed employs large language models (LLMs) for task synthesis and filtering, trajectory generation, and trajectory filtering, we emphasize that all models used in our work are open-source and can be deployed efficiently using vLLM, which significantly reduce inference overhead. Our implementation demonstrates that high-quality, end-to-end synthetic data generation is feasible without reliance on expensive proprietary models.
>
> Research groups with extremetly limited computational resources can nearly reproduce results using the `MistralAI` family models for task synthesis and `GPT-OSS-20B` for trajectory generation and filtering, which is already supported in our codebase shared in our supplementary materials. Furthermore, if the goal is onboarding new MCP servers to augment domain coverage, researchers could use resource-efficient model alternatives fully compatible with our pipeline. We share GPU requirements for the original models used to build Toucan, as well as recommended model alternatives in `Appendix E`.
>
> Finally, we emphasize that a core goal of this work is to build open, high-quality, and permissively licensed tool-agentic data resources for the research community. With Toucan, researchers no longer need to manually collect MCP metadata and generate diverse trajectories from scratch, significantly lowering the barrier to developing and training more capable agentic LLMs.
>
> ---
>
> > **Comment 2:** Models trained on this dataset may be overfitted to the 2000 tools. How to extend the models to other domains with different tools? These paper may have limited usefulness to practical tasks.
>
> In response to the reviewer’s concern about potential overfitting to the ~2,000 tools in our dataset, we want to clarify that our dataset design choices and evaluation benchmarks directly address and mitigate this risk.
>
> First, the Toucan dataset spans 27 domains (Figure 3), substantially reducing the risk of domain-specific memorization. Second, our evaluation includes MCP Universe, where four out of its six domains (i.e., location navigation, repository management, 3D design, and web search) are entirely out-of-distribution (OOD) relative to our training data, as these tools require (paid) APIs and are therefore excluded from Toucan. The strong performance in Toucan-tuned models in these tasks demonstrates that Toucan can successfully generalize to out-of-distribution (OOD) domains. We explicitly demonstrate this domain overlap analysis in `Appendix B.4` of the revised manuscript.
>
> Overall, the broad domain coverage of our dataset and good performance in OOD domains demonstrate that our models learn generalizable tool-use behavior rather than memorizing specific tools.
>
> [Continued]

---

> > ### Author Response · Authors · 2025-11-21
> > **Response for Reviewer Rknv [Part2]**
> >
> > > **Comment 3:** The pipeline described for task generation and filtering is similar to previously proposed methods, such as in [a] (Multi-modal Agent Tuning: Building a VLM-Driven Agent for Efficient Tool Usage, ICLR 2025). While the data volume is significantly larger in this manuscript, the methodological novelty may be questioned. The relationship and differences should be discussed in the manuscript.
> >
> > We thank the reviewer for pointing out MM-Traj (Gao et al., 2025) as a relevant omission in our related work. In response to the comment regarding methodological novelty, we want to clarify that MM-Traj and Toucan differ substantially in their research goals, task synthesis approach, task and trajectory filtering rigor, reliance on frontier models, dataset expansion strategy, and overall scale.
> >
> > * **Research goal.** MM-Traj is designed specifically for multimodal tool use and focuses on VLM-driven agents solving tasks that involve images and structured visual inputs. Toucan instead targets text-based tool use, covering 27 diverse functional domains. Thus, while both datasets are tool-agentic, they address different research goals.
> > * **Task synthesis strategy.** MM-Traj generates tasks through a prompt-engineering ensemble. In contrast, Toucan synthesizes tasks from real MCP server files by leveraging the generation strategies described in `Section 3.1`.
> > * **Filtering rigor.** Toucan integrates rule-based verifiers and LLM judges at both the task and trajectory levels, ensuring quality, correctness, novelty, and consistency. MM-Traj applies a more limited filtering procedure (primarily for image generation).
> > * **Frontier model dependency.** MM-Traj relies on closed-source models (e.g., GPT-4o mini) for synthetic task and input-image generation. In contrast, our pipeline demonstrates that end-to-end SFT trajectories can be generated with open-source LLMs, making the pipeline more accessible to the open-source community.
> > * **Dataset expansion.** Because MM-Traj depends on new frontier LLMs for high-quality generation, dataset expansion is coupled to the release of new frontier models. Toucan’s extensibility, however, is enabled by MCP onboarding: generating tasks for new domains requires of new MCP specification files and their respective environments.
> > * **Size and diversity.** MM-Traj contains ~20k trajectories, appropriate for its multimodal setting. Toucan, by contrast, produces 1.5M+ trajectories across a large and diverse domain space.
> >
> > We have added the MM-traj dataset to our related work section.
> >
> > > **Comment 4:** The dataset involves more than 2,000 tools. It is unclear how the inclusion of so many tools adds unique value to the dataset. A discussion of the redundancy and actual diversity of these tools would help clarify whether they introduce meaningful variety or if many of them are effectively duplicates.
> >
> > We appreciate the reviewer’s question regarding the added value of including more than 2,000 tools. Our design choice is motivated by the observation that tool diversity is a key driver of agent robustness, and our dataset construction process explicitly ensures that the resulting tasks are not redundant or near-duplicate variants of one another regardless of which tools they require.
> >
> > To further investigate to what extend tool diversity benefits models' performance in tool-agentic tasks, we have conducted an additional ablations (see below and also in `Appendix C.4` in the revised paper). In this experiment, we create five subsets of Toucan datasets, each doubling the number of tools relative to the previous one. The number of tools preserved in the dataset ranges from 100 to 1,600, while the total number of trajectories is kept constant at 20,000. We fine tuned `Qwen2.5-32B-Instruct` on these subsets, and evaluated the resulting models on BFCL V3 and  $\tau$-Bench. Our results show a **consistent upward trend in overall performance as tool diversity increases**, indicating that **a larger and more diverse tool set leads to better generalization** rather than redundant learning.
> >
> > **Tool Diversity Ablation on BFCL V3 and  $\tau$-Bench**
> >
> > | # Tools | Model Variant | BFCL V3 Overall | Non-Live AST | Live AST | Multi-Turn | Relevance | Irrelevance |  $\tau$ Airline Pass@1 |  $\tau$ Retail Pass@1 |
> > |----:|----|----|----|----|-----|-----|----|-----|---|
> > | 100  | Qwen2.5-32B-Instruct-Toucan-100Tools-20K  | 60.38% | 87.58% | 64.82% | 36.00% | 88.89% | 46.66% | 28.50% | 47.39% |
> > | 200  | Qwen2.5-32B-Instruct-Toucan-200Tools-20K  | 60.90% | 86.56% | 65.44% | 37.00% | 88.89% | 49.50% | 27.50% | 50.43% |
> > | 400  | Qwen2.5-32B-Instruct-Toucan-400Tools-20K  | 61.99% | 87.48% | 65.08% | 40.00% | 83.33% | 49.00% | 28.50% | 48.48% |
> > | 800  | Qwen2.5-32B-Instruct-Toucan-800Tools-20K  | 61.73% | 87.31% | 64.73% | 40.38% | 83.33% | 45.98% | 29.25% | 50.54% |
> > | 1600 | Qwen2.5-32B-Instruct-Toucan-1600Tools-20K | **62.26%** | 86.27% | **67.57%** | 39.38% | 83.33% | **52.08%** | **29.75%** | **52.06%** |

---

> ### Author Response · Authors · 2025-11-25
> **Follow-Up on Author Response**
>
> Dear reviewer,
>
> I am writing to follow up on our author response submitted on November 21st. With the December 2nd revision deadline approaching, we are concerned that we may not have enough time to address any additional questions you might have.
>
> We would greatly appreciate your engagement in the discussion at your earliest convenience.
>
> Thank you for your time and efforts. We look forward to hearing from you.

---

> > ### Author Response · Authors · 2025-11-27
> >
> > Dear Reviewer,
> >
> > I hope you are doing well. I am writing again to kindly follow up on our previous message, as we have not yet received a response. With the revision deadline approaching, we want to ensure we have sufficient time to incorporate any further feedback you may have.
> >
> > Thanks in advance,

---

### Author Response · Authors · 2025-12-02
**Summary of Author-Reviewer Discussion**

Dear Reviewers and Area Chairs,

Thank you very much for your time in reviewing our paper and for the insightful feedback and comments. Below, we summarize our responses to the reviewers’ concerns and highlight the additional experiments we conducted.

---

## Summary of Strengths Mentioned by Reviewers

**Huge Impact to Open Source Community**: Toucan is the largest open-source tool-agentic dataset, providing large-scale, high-quality, and diverse real-world tool-agentic interactions. It fills a significant gap in available open-source datasets for training agentic LLMs (`Rknv`, `tfUW`, `CckG`, `uq6F`).

**Carefully Designed Modular Synthetic Data Pipeline**: The five-stage pipeline with rigorous filtering (LLM-based quality assessment and rule-based validation) and three thoughtful extensions (irrelevance handling, persona-based diversification, and multi-turn conversations) demonstrates careful engineering. The ablation study validates the contribution of each component (`tfUW`, `CckG`). The modular pipeline design allows future extensions and customization (`tfUW`).

**Strong Performance**: Fine-tuning on Toucan leads to clear performance improvements, with our models outperforming existing baselines on multiple benchmarks (`Rknv`, `uq6F`).

**Wide Domain Coverage**: Toucan covers a wide variety of domains, tools, and interaction patterns (`Rknv`).

**Detailed Analysis**: The analysis shows the diversity and quality of the dataset based on LLM judges (`uq6F`).

**Clear Writting**: The paper is well-organized and easy to follow, with clear presentation of dataset statistics, filtering criteria, and evaluation metrics (`CckG`).

---

## Highlights of Our Clarification and New Experiments

**Extended Evaluation Results:** We perform a series of evaluations and ablations to better characterize Toucan’s impact. Specifically, we (i) conduct tool-diversity ablations by varying the number of tools (100–1,600) while keeping trajectories fixed, showing a consistent performance gain as tool coverage increases; (ii) run data-scale ablations (20K–119K trajectories), revealing steady improvements; and (iii) ablate the final rule & LLM-based filtering stage, demonstrating that the full five-stage pipeline yields stronger BFCL performance and reduced hallucinations compared to unfiltered data. We additionally compare training on the Toucan SFT subset versus the full 1.5M dataset, showing that our rebalanced mixture remains competitive while being substantially more compute-efficient. All results are reported in the revised appendices.

**Justification about Toucan on OOD setups:** We clarify that all τ-Bench / τ²-Bench tools are excluded from Toucan’s training data, and four MCP Universe domains (location navigation, repository management, 3D design, web search) are entirely out-of-distribution because the corresponding servers require paid/authenticated APIs. Toucan-tuned models still show strong gains in these settings, supporting that they learn generalizable tool-use behaviors rather than memorizing specific tools or domains (Appendix B.4).

**Cross-model-family validation beyond Qwen:** We also fine-tune Llama-3.1-8B-Instruct and Llama-3.3-70B-Instruct on Toucan; both show large, consistent gains on all BFCL benchmark subsets, confirming that Toucan transfers well across architectures and scales (Table 2).

**Comparison with existing public tool datasets:** At similar data scale and with the same base model (Qwen2.5-14B-Instruct), Toucan matches Nemotron-SFT on BFCL overall but clearly outperforms it on τ-Bench and τ²-Bench, indicating stronger multi-tool, cross-domain agentic reasoning.

**Additional analyses and clarifications:** In addition, we (i) report inter-annotator agreement for two humans and LLM judges, showing LLM-as-a-judge is close to the human agreement ceiling; (ii) add more qualitative Toucan trajectory examples (multi-tool, multi-step, error recovery); and (iii) more clearly state Toucan’s novelty as an MCP-native, extensible, open-source pipeline for large-scale tool-agentic data.

---

## Conclusion

We sincerely thank all reviewers for their thoughtful feedback. Our responses and new experiments show that Toucan is effective, high-quality, and makes a substantial contribution to open-source agentic research.

We are committed to incorporating the reviewers’ insights to further strengthen the paper. However, due to the recent data leakage incident on OpenReview, we are unable to engage in further author–reviewer discussion to address remaining concerns (particularly those from reviewer `Rknv`), which may affect the overall assessment. We respectfully ask that the AC take our comprehensive responses and new results into full consideration when making the final decision.

---

### Meta-Review · Area_Chair_NQLL · 2026-01-18

**Summary:**

The paper introduces TOUCAN, a large-scale tool-agentic dataset consisting of 1.5M trajectories synthesized from nearly 500 real-world MCP servers. The work presents a multi-stage data generation pipeline with task synthesis, trajectory generation, and rule- and LLM-based filtering, along with extensions for irrelevance handling, persona diversification, and multi-turn interactions. Experiments show that models fine-tuned on TOUCAN improve performance on a number of benchmarks.

Reviewers broadly agreed that the dataset is large, well engineered, and potentially valuable to the open-source community. However, there were persistent concerns about the technical novelty of the pipeline, the strength and interpretation of the experimental evidence, and whether the contribution rises beyond a large-scale engineering effort.

**Reviewer Concerns:**

A central concern across the reviews is limited technical novelty. Reviewers noted that core components of the pipeline, including LLM-based task synthesis, rule- and LLM-based filtering, and synthetic trajectory generation, closely resemble prior tool-agentic datasets such as ToolACE, APIGen, and MM-Traj. While the authors argue that operating over real-world MCP environments is a meaningful step beyond fixed APIs, reviewers largely characterized this as an engineering extension in scale and infrastructure rather than a fundamentally new methodological contribution.

Reviewers also raised concerns about the experimental justification of scale and quality. Several reviews questioned why training and evaluation were conducted on a 119K subset instead of the full 1.5M trajectories, how the reported gains should be interpreted relative to strong baselines, and whether the improvements sufficiently justify the dataset scale. Although the rebuttal adds data-scale ablations, cross-model evaluations, and additional comparisons, these additions do not fully resolve concerns about effectiveness and generality.

Another recurring concern involves generalization and evaluation reliability. Reviewers questioned potential overlap between training data and benchmark tools, the reliance on LLM-based judges given reported human agreement levels, and the exclusion of authenticated MCP servers, which may limit coverage of realistic workflows. While the rebuttal provides clarifications and additional analyses, these issues remain points of uncertainty in the current submission.

Overall, the concerns focus on novelty, empirical justification, and generalization, and reflect structural limitations rather than missing clarification.

**Reviewer Scores:**

- Reviewer Rknv (original score: 4) would likely remain around the same score. Their concerns center on limited novelty, computational cost, and the risk of overfitting to a large set of tools, which are only partially addressed by the rebuttal.
- Reviewer tfUW (original score: 6) would likely remain a similar score. While the reviewer acknowledged the value of the dataset, they raised persistent concerns about evaluation reliability, authenticated server exclusion, and generalization.
- Reviewer CckG (original score: 6) explicitly stated after the rebuttal that they were keeping their original rating.
- Reviewer uq6F (original score: 2, updated to 4) already revised their score upward after additional experiments were provided. However, even after this increase, the reviewer maintained that the contribution is primarily an engineering effort rather than a substantive research advance. A further increase beyond the revised score is unlikely.

Overall, while the rebuttal improves clarity and soundness, the expected score changes do not indicate a shift toward a clear acceptance consensus.

---

### Decision · Program_Chairs · 2026-01-26

Reject